# Deep learning to estimate lithium-ion battery state of health without additional degradation experiments

Jiahuan Lu[1], Rui Xiong ![ORCID][1] ✉, Jinpeng Tian ![ORCID][1] ✉, Chenxu Wang[1] & Fengchun Sun[1]

State of health is a critical state which evaluates the degradation level of batteries. However, it cannot be measured directly but requires estimation. While accurate state of health estimation has progressed markedly, the time- and resource-consuming degradation experiments to generate target battery labels hinder the development of state of health estimation methods. In this article, we design a deep-learning framework to enable the estimation of battery state of health in the absence of target battery labels. This framework integrates a swarm of deep neural networks equipped with domain adaptation to produce accurate estimation. We employ 65 commercial batteries from 5 different manufacturers to generate 71,588 samples for cross-validation. The validation results indicate that the proposed framework can ensure absolute errors of less than 3% for 89.4% of samples (less than 5% for 98.9% of samples), with a maximum absolute error of less than 8.87% in the absence of target labels. This work emphasizes the power of deep learning in precluding degradation experiments and highlights the promise of rapid development of battery management algorithms for new-generation batteries using only previous experimental data.

Lithium-ion batteries (LIBs) offer high energy density, fast response, and environmental friendliness[1], and have unprecedentedly spurred the penetration of renewable energy[2–4]. The global market of LIBs displays staggering figures in 2020, up to 142.8 GWh on the side of electric vehicles, and it is expected to exceed 91.8 billion dollars[5] in the next few years. While LIBs are being popularized at a phenomenal rate, their prolonged applications are facing tough challenges. As is the case with machines, LIB components such as electrodes and separators experience varying levels of degradation. These negative spillovers lead to capacity and power fade[6,7] and thereby imperil the assets[8]. To ensure safe and efficient battery management, obtaining an accurate battery state of health (SOH) is of vital importance.

Battery SOH has been defined in various forms. It can be defined by the service time[9] or by the increase in the internal resistance[10]. Although these variables are easily measurable, battery degradation is also accompanied by capacity loss, whose accurate determination

impacts other battery management tasks such as driving range estimation and life prediction. Thus, SOH, defined as the ratio between the present capacity and the initial capacity, is drawing broad attention[11,12]. However, the capacity measurement requires completely charging or discharging the batteries with specific protocols[13], which is not practical for batteries in use. This motivates the SOH estimation from daily operating data.

Existing SOH estimation studies are generally devoted to extracting features correlated with SOH degradation and mapping them to the SOH. These methods require lifelong battery degradation data with measured SOH labels of the target LIBs (so-called target-labeled data). On this basis, many features have been crafted based on our understanding of battery degradation, such as the electrical features[14,15], electrochemical features[16], acoustic features[17], mechanical features[18,19], and thermal features[20]. Such arduous data collection and feature engineering steps impede the development of SOH estimation

[1]Department of Vehicle Engineering, School of Mechanical Engineering, Beijing Institute of Technology, Beijing 100081, China. ✉e-mail: rxiong@bit.edu.cn; tianjinpeng@bit.edu.cn

methods. Recently, deep neural networks (DNNs)[21,22] have demonstrated the automatic extraction of black-box features from raw operating data, showing impressive SOH estimation performance. However, experimental collection of the target-labeled data is time-consuming and resource-intensive[23], and creating massive datasets for different types of LIBs is rarely sustainable.

A paradigm of generalization for DNNs is transfer learning, which is one promising solution to lighten the burden of data collection. Transfer learning transfers knowledge learned by a DNN in a training dataset (the source domain) to a different dataset (the target domain) using a small number of target-labeled data[24]. For example, a DNN for charging curve prediction trained using one type of battery can adapt to other types of batteries by fine-tuning using a small number of new samples[25]. A growing body of literature[26,27] applies retraining or fine-tuning techniques for SOH estimation of various types of batteries. Generally, these works require at least 25–30% of labeled lifelong data in the target domain. Therefore, target domain degradation experiments are still needed. These approaches relying on conventional experiments cannot keep pace with the battery upgrading and have established a barrier to the development of battery technologies.

Approaches without the need for additional target-labeled data to estimate SOH are attractive[28–30]. This can help the rapid development of battery management systems (BMSs) for new-generation batteries using only existing experimental data, saving considerable time and resources. It is also expected to motivate the utilization of large-scale field data without labels. Indeed, issues for target label-agnostic cases have long been noticed. Cross-domain learning in the absence of target labels has been proven to be equivalent to a dual training task, i.e., learning to predict the source labels while closing the gap between the source and target domains[31,32]. Research in the field of visual recognition[33] has shown that even though there are no target-labeled data for training, a DNN jointly trained for classification and domain invariance can conduct classification precisely across visually distinct domains. This confirms that DNNs can accomplish the tasks in target label-agnostic cases. However, LIB SOH estimation, which has a massive demand for target labels, has yet to benefit from it.

In this work, we propose a deep learning-based framework to estimate battery SOH without relying on target labels for training. The proposed framework integrates the estimates of a swarm of DNNs into a reliable SOH estimate rather than relying on a single DNN. Individual DNNs are trained to learn cross-domain knowledge according to source labels and domain invariance of degradation features. DNNs with good performance in the swarm are selected for reliable estimation. We further reveal the influence of the sample distribution in the source domain on SOH estimation and propose to improve estimation performance by trimming the sample distribution in the source domain. We adopt two self-developed and three public battery degradation datasets for cross-validation. The validation covers 80 cases, encompassing 71,588 samples collected from 65 cells. We demonstrate that the proposed framework can achieve an absolute error of less than 3% for 89.4% of samples (less than 5% for 98.9% of samples), with a maximum absolute error of less than 8.87%. To provide references for the selection of hyper-parameters, we also investigate the influence of the crucial hyper-parameters on the estimation performance. These results highlight the potential of deep learning in supplanting the time-consuming battery degradation experiments, and further rapid development of BMS for new-generation batteries using existing experimental data.

## Results
### Framework overview
We develop a SOH estimation framework composed of a swarm of DNNs (Fig. 1). This framework is designed for reliable estimation by selectively integrating the estimates from multiple DNNs. The proposed framework is introduced in terms of the training procedure

(Fig. 1a), estimation procedure (Fig. 1c), and its component units (Fig. 1b). Their definitions and processes are introduced in the Methods section in detail.

The training procedure of the proposed framework integrates independent sub-trainings of $N$ DNNs (see Fig. 1a). Without loss of generality, the battery charging data are employed as the input of the DNNs since the battery charging process is generally controllable and occurs regularly. Specifically, charging capacity sequences within a voltage sampling window (so-called partial charging curves) are taken as the input of each DNN, as demonstrated by previous studies[25,34–37]. Before sub-trainings, partial charging curves from both source and target domains are normalized by their nominal capacity. When the training starts, all the sub-trainings are enabled and share an identical training set that is composed of labeled source domain samples and unlabeled target domain samples. We also designed a trimming round to form a new source domain with a balanced SOH distribution by discarding some samples. The training procedure of the proposed framework is terminated after all the sub-trainings are finished.

Each DNN in the proposed framework has identical hyper-parameters (Fig. 1b) but is initialized with different random seeds based on the He initializer[38]. As the input of DNN, partial charging curves from both source and target domains are first gridded with a voltage interval of 10 mV to reduce the data burden. Next, these samples are fed into stacked one-dimensional (1D) CNN layers to extract their feature vectors. After that, feature vectors of the source domain are flattened and fed into a terminal fully connected (TFC) layer to generate their SOH estimates. These estimates are used together with the source domain labels to calculate the source domain loss. On the other hand, feature vectors of target-domain samples are flattened to a middle fully connected (MFC) layer for reconstructing their feature vectors. These reconstructed feature vectors play two roles. The first is to quantify the domain gap together with the source domain feature vectors. The second is to provide estimates of target domain samples (treated as the pre-estimates of each trained DNN) in the estimation procedure, where the reconstructed feature vectors are further fed into the same TFC as the source domain for regression. By simultaneously minimizing the SOH estimation loss of source domain samples and the gap between the TFC inputs of the two domains, each sub-training transfers the source domain knowledge to the target domain.

The estimation procedure of the proposed framework, unlike the training procedure, is to select a swarm of the trained DNNs to participate in the estimation (Fig. 1c). First, all the DNNs are activated to estimate the SOHs in the target domain, as mentioned above. The trained DNNs are expected to differ widely in estimation performance owing to the training uncertainty and can thus be treated as pre-estimators. To produce a reliable final estimate, we eliminate some unfavorable DNNs by setting quartile thresholds for the mean and standard deviation of the estimation results. The estimations from the selected DNNs are averaged to produce eventual SOH estimates for the target domain samples. Detailed discussions can be found in the Rationalization of predictive performance section.

### Data generation
SOH estimation in target label-agnostic cases spans different applications, manufacturers, and chemistries. To reflect such situations, we employ 10,757 samples collected from 65 commercial LIB cells produced by five different manufacturers for validation. The eventual datasets cover five kinds of widely-used cathode active materials, including lithium cobalt oxide ($LiCoO_2$, LCO)[39], a blend of LCO and lithium nickel cobalt oxide ($LiCoNiO_2$, LCO/NCO)[7], nickel manganese cobalt ($Li(NiMnCo)O_2$, NMC)[40], nickel cobalt aluminum ($LiNiCoAlO_2$, NCA), and lithium iron phosphate ($LiFePO_4$, LFP). Note that the exact composition of the cathode active materials cannot be further

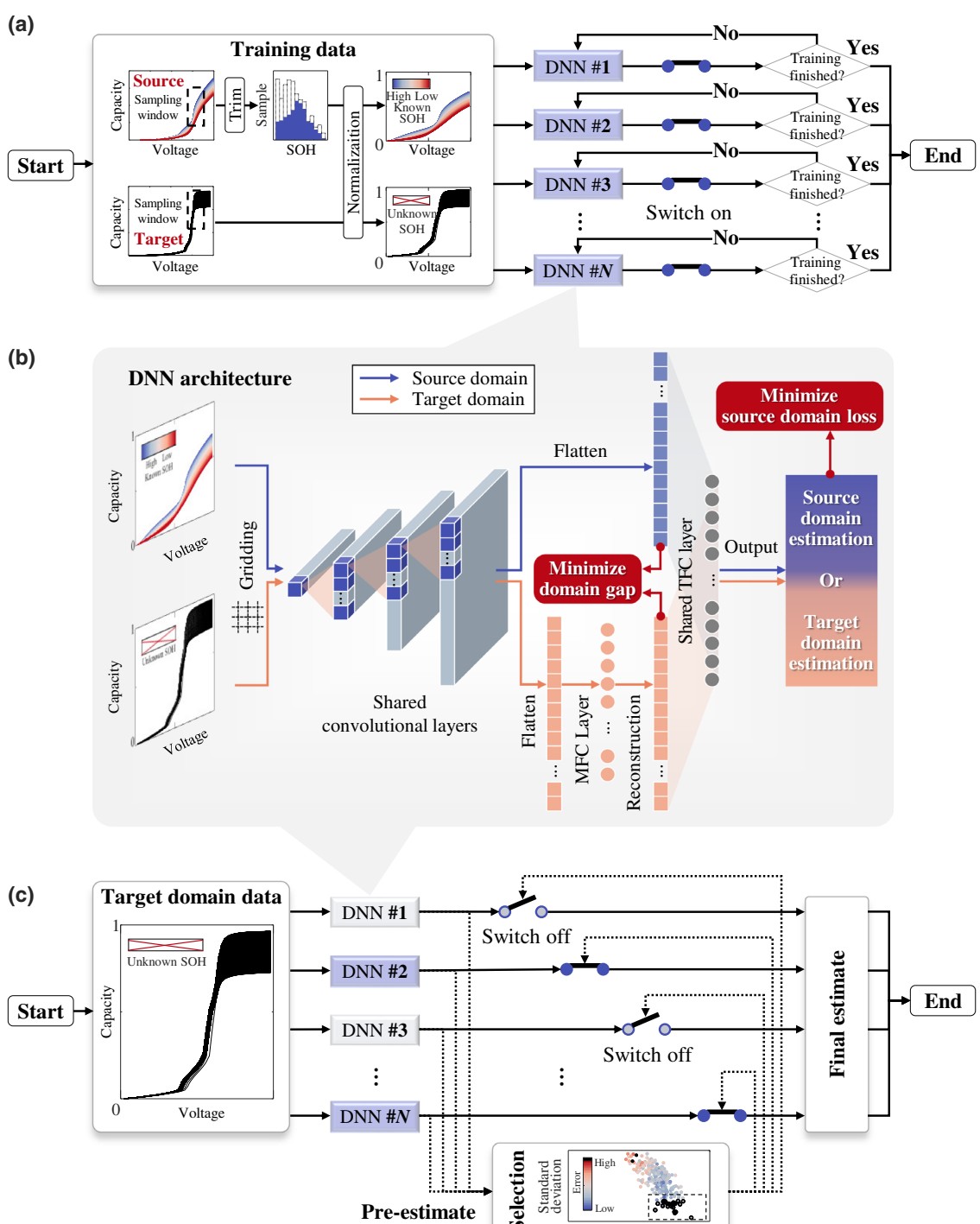

**Fig. 1 | Overview of the proposed SOH estimation framework. a** Diagram of the training procedure. **b** Architecture of each DNN of the swarm. **c** Diagram of the estimation procedure.

provided here as this information is not available in the existing literature. The specifications of these five types of LIBs and their experimental costs are compared in Table 1 (see Supplementary Note 1 for the estimation of the experimental cost). The degradation data of PANASONIC and GOTION LIBs are experimentally generated in our lab (see Data availability statement), and others are from three public datasets[39,41–43]. Details regarding the datasets can be found in Table S1. Note that C-rate is a measure of the battery's charge or discharge current relative to its nominal capacity, and is used here to describe the experimental current.

In Fig. 2, we plot the charging curves of the selected LIBs to disclose their different degradation behaviors. Significant gaps exist between the charging curves of any two types of LIBs, even though Datasets #2 and #4 consist of batteries with similar electrode active materials. This is because the degradation behavior of LIBs is subject to various factors, such as manufacturing factors and application scenarios. Affected by dissimilarity among experiments, degradation rate, and data processing, the LIBs we employed cover different SOH distributions. This simulates the real-world discrepancy in sample distribution between the source and target domains. Therefore, given a

**Table 1 | Main specifications of the selected LIBs in this work**

| Dataset | Manufacturer/ provider | Electrode active materials (Cathode/Anode) | Nominal capacity (Ah) | Voltage range (V) | Data amount (Samples) | Estimated test duration (Hours) |
|---------|------------------------|---------------------------------------------|------------------------|--------------------|------------------------|----------------------------------|
| #1 | CALCE[39,43] | LCO/ Graphite | 1.1 | 2.7-4.2 | 2807 | 1397 |
| #2 | SANYO[41] | NMC/ Graphite | 1.85 | 3.0-4.1 | 415 | 644 |
| #3 | PANASONIC | NCA/ Graphite | 3.03 | 2.5-4.2 | 2770 | 1801 |
| #4 | KOKAM[42] | (LCO/NCO) / Graphite | 0.74 | 2.7-4.2 | 503 | 8473 |
| #5 | GOTION HIGH-TECH | LFP/ Graphite | 27 | 2.0-3.65 | 4262 | 2238 |

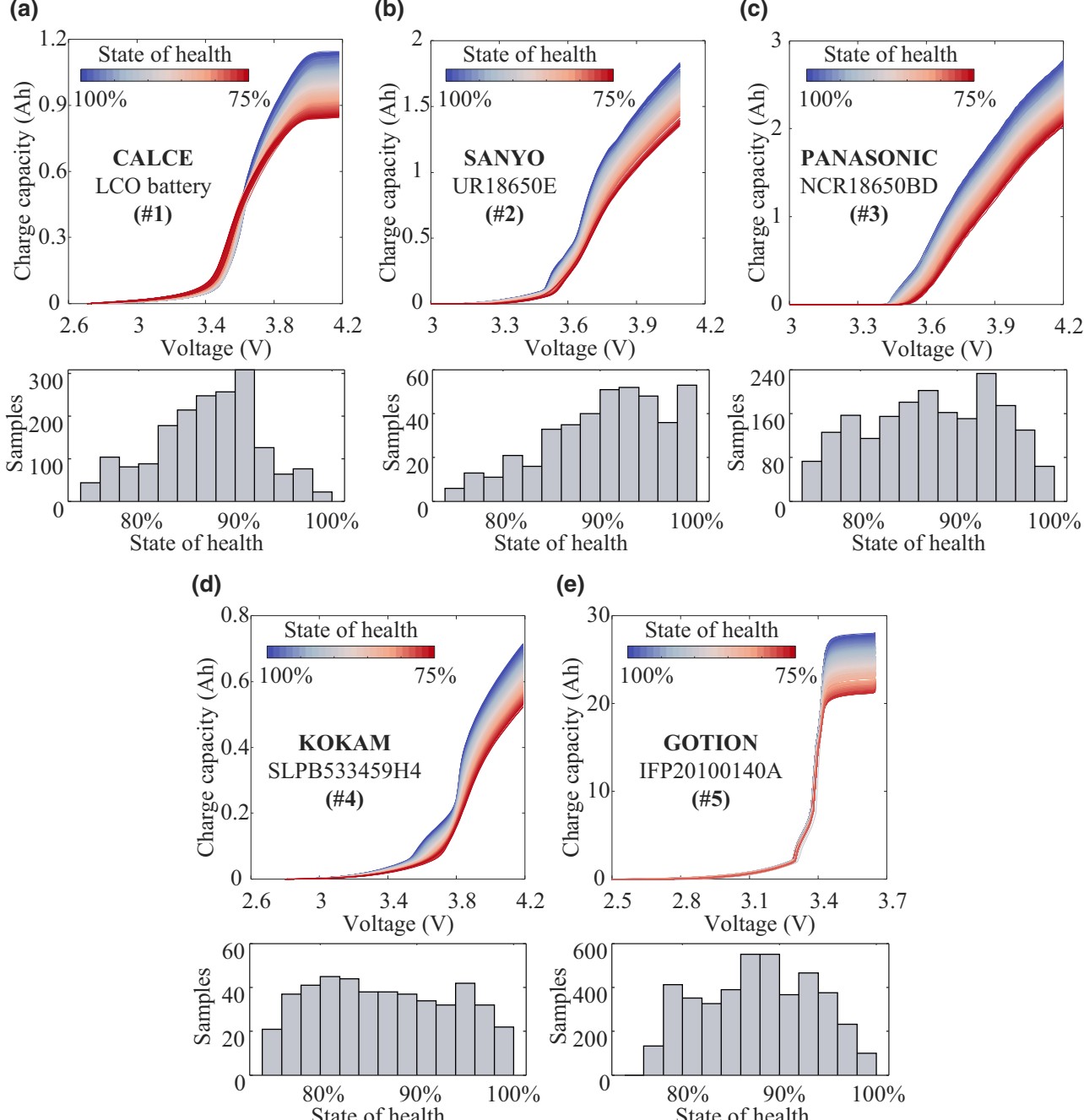

**Fig. 2 | Lifelong charging curve family and histogram of the SOH of the selected five types of LIBs produced by five manufacturers. a** #1, **b** #2, **c** #3, **d** #4, **e** #5. The histograms in subplots show the number of samples with different SOH.

new type of LIBs, traditionally, we carry out time- and resource-consuming experiments to develop tailor-made models for SOH estimation. To address this issue, we propose a domain adaptation-enabled framework for cross-dataset battery SOH estimation without knowing any SOH labels of target batteries.

## Cross-dataset battery SOH estimation in the absence of target labels

In this experiment, we examine the performance of the proposed framework on cross-dataset SOH estimation in the absence of target labels. Detailed hyper-parameter settings can be found in Fig. S1. To this end, the employed five types of LIBs are pairwise combined, resulting in a total of 20 combinations. A general case is that there are lifelong labeled source domain data (SOH ranges from 100 to 75%) and unlabeled target domain data (SOH ranges from 100% to an unknown level) for training. That is to say, except that initial cycles have a SOH of 100%, we know nothing about the SOH distribution of samples from the target domain. Thus, our estimation faces the challenge of domain imbalance[44]. To imitate general cases, we postulate that the SOH of batteries in the target domain are distributed from 100 to 95, 90, 85, and 80% (denoted as -95, -90, -85, and -80%), respectively. As a result, each combination contains four cases with four different lower SOH bounds (see Table S2 for the detailed validation scheme). A 500-mV voltage window is utilized to extract the partial charging data, and it covers 3.5 to 4 V for Datasets #1–4 and 3.1 to 3.6 V for Dataset #5. The impact of different voltage windows on SOH is investigated in Supplementary Note 2.

We first use all source domain samples to train the framework without trimming and the estimation of SOH error distribution is shown in Fig. 3a. Detailed results can be found in Figs. S3, 4. Overall, the mean absolute error (MAE) of the proposed framework in all cases is within 5.01%, which proves that the estimation in the absence of target labels is effective. Interestingly, changing source domains induces variation in the estimation accuracy for a specific target domain. For example, using Dataset #5 as the source domain, we can achieve accurate estimation for Datasets #1–4. However, using Dataset #2 as the source domain leads to different estimation performance: the overall errors in the -85 and -80% cases are generally higher than those in the -95 and -90% cases. This result draws our attention to the source domain. Tracing back to the SOH distribution in the source domains, we find that the SOH distribution in Dataset #2 is significantly skewed towards high SOH, and the skewness is −0.4. In contrast, the distribution of other datasets is relatively symmetric, and the skewnesses of Datasets #1 and #3–5 are −0.19 and −0.12, 0.12, and −0.05, respectively. This explains why using Dataset #2 as the source domain brings significant advantages in the -95 and -90% cases. For further validation, we trim the source domain SOH distribution to be symmetric by making the skewness tend towards zero before training. The SOH distributions in source domains after the trim are shown at the top of Fig. 3a. Dataset #5 undergoes only slight trimming as its original distribution is almost symmetric (the skewness before the trim is −0.05). In contrast, Dataset #2 is significantly trimmed, where many samples with high SOH are discarded. Next, we employ the trimmed source domains to train the framework and evaluate the performance of SOH estimation. On the whole, the extent of change in accuracy is positively correlated with that in the trim. Using Dataset #5 as the source domain leads to good performance with little change as before. Using Dataset #2, after undergoing the most notable change, brings similar trends as the other datasets: the framework performs significantly better in the -95 and -90% cases than in the -85 and -80% cases. These results highlight the impact of source domain SOH distribution on SOH estimation in the absence of target labels and the effectiveness of the trim. We then gather all the verification cases to statistically evaluate the improvement of SOH estimation. Figure 3b shows the comparative results before and after the trim to describe the error distribution as a

function of true SOH. Trimming the source domains reduces the maximum absolute error from 10.09 to 8.87%. More importantly, the percentage of high absolute errors (>5%) is dramatically reduced. Also, most estimates are at a low absolute error level (≤3%). To quantify this, we plot the cumulative distribution of the absolute error using bins with a width of 1% absolute error in Fig. 3c. 89.4% of the samples have an absolute error of less than 3% and up to 98.9% of the samples have an error of less than 5% after the trim, which is significantly superior to the case without trim. In conclusion, the proposed framework can achieve accurate cross-dataset SOH estimation in the absence of target labels and can be improved after trimming the sample distribution in the source domain.

## Comparison with existing methods

To verify the advancement of the proposed framework, we gather all validation cases and compare our accuracy with that of four popular methods, including Gaussian process regression[35] (GPR), random forest[45] (RF), support vector regression[46] (SVR), and CNN[47]. Their hyper-parameter settings can be found in Table S3. The comparative results of the absolute error distribution are described in Fig. 4, and detailed results can be found in Figs. S5–8.

We first show the performance of the four existing methods when the target domain labels are available. Having enough target labels for learning the target domain, the existing methods show high accuracy with MAEs of less than 1%. However, the target labels in practice come at the cost of numerous workforce and energy. Developing a battery degradation dataset requires 644–8473 hours of degradation experiments (according to the estimation in Supplementary Note 1). In the absence of the target labels, existing methods fail to provide reliable estimation with their MAEs over 5.01%, and the maximum absolute error reaches over 17.91%. By contrast, the proposed framework achieves accurate SOH estimation without target labels, reducing the MAE and maximum absolute error by more than half. The MAE and maximum absolute error are within 1.43 and 8.87%, respectively. More importantly, given a swarm size of 300, our method leverages -0.7 hours for training, avoiding degradation experiments of thousands of hours (see Supplementary Note 3 for the computational cost comparison). This excellent performance can be attributed to the swarm-driven and domain adaptation strategies. To demonstrate this, ablation experiments are performed to verify the role of these strategies. Benchmark 1 and Benchmark 2 are created by disabling the swarm-driven and domain adaptation strategies of the proposed framework, respectively. Benchmark 3 is designed by disabling both strategies. The detailed results can be found in Figs. S9–11. Besides, Benchmark 1 is designed with a comparable number of hyper-parameters to the proposed framework. As expected, without the help of either of the two strategies, the estimation performance approximately reduces to the level of existing methods.

## Rationalization of predictive performance

The excellent performance of our framework can be attributed to domain adaptation and swarm-driven strategies. The swarm-driven strategy is first analyzed. We take a pair of instances in the -85% case (i.e., the cases of transfer from Dataset #1 to #5 and from Dataset #5 to #1) to investigate its influence. The distributions of pre-estimation root mean square errors (RMSEs) of DNN swarms in these cases are reported in Fig. 5a, b. Overall, the pre-estimation RMSEs of DNN swarms before selection, which are affected by uncertain training, have a wide distribution. One can note that some DNNs have RMSEs of up to 10%. Thus, relying only on a single DNN may yield unreliable SOH estimates like this. We also observe that most DNNs in the swarm are positively skewed within an RMSE of less than 8%. Many of them have RMSEs of less than 3%, indicating that a considerable part of the swarm is trustworthy. This motivates the proposed framework to selectively integrate estimations of a swarm of DNNs for reliable SOH estimation.

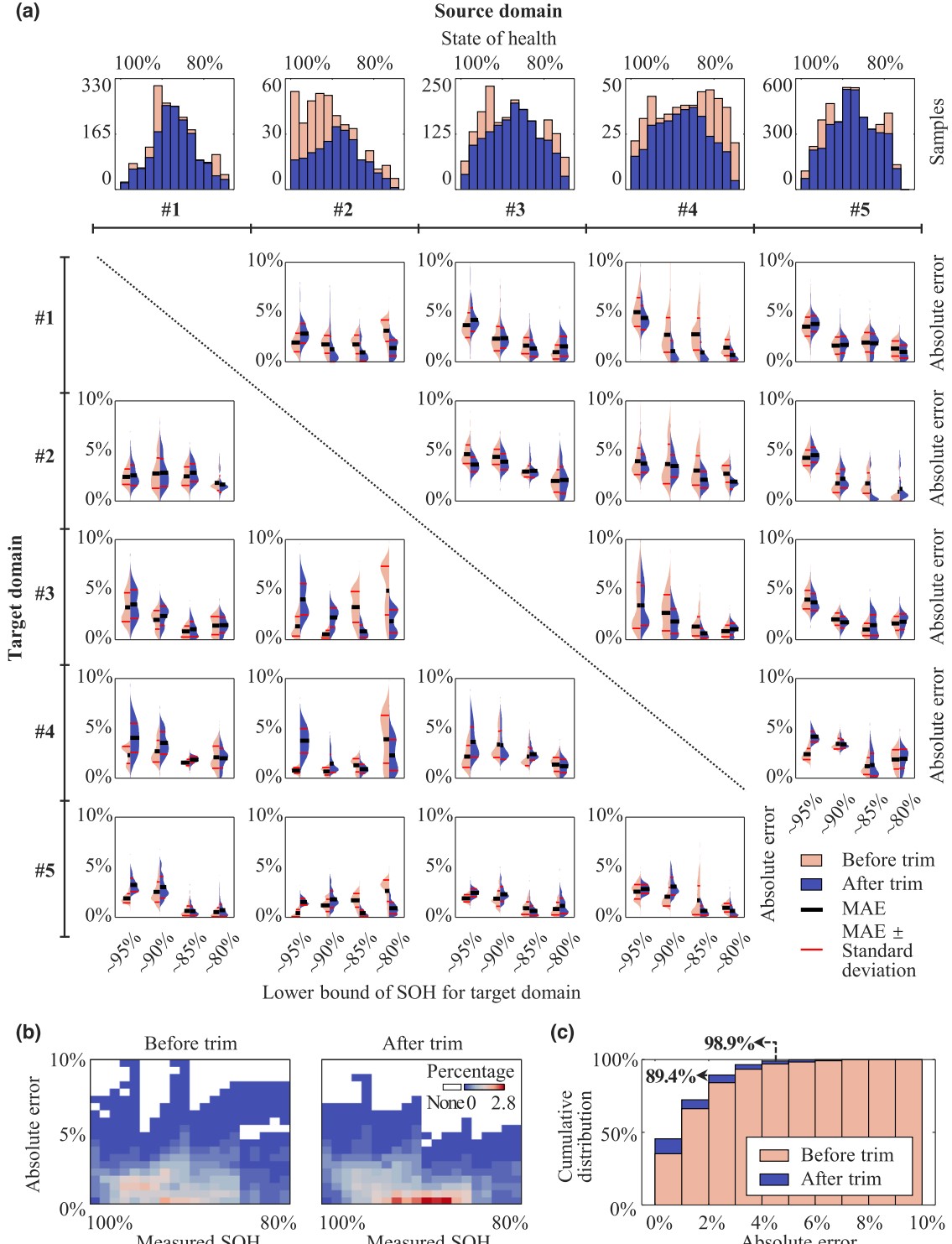

**Fig. 3 | Performance of cross-dataset battery SOH estimation in the absence of target labels. a** Distribution of absolute error versus target domain SOH range. **b** Distribution of absolute error of the proposed framework as a function of measured SOH. Both plots use the same color scales. **c** The cumulative distribution of estimation absolute errors.

We find that, for a given batch of target domain samples, the estimation results of the DNNs in the swarm are diverse, but their distributions show regularity with their accuracy (see Supplementary Note 4 for the analysis). This provides an opportunity for the proposed framework to select a group of well-performing DNNs. To develop a selection criterion, we choose the mean and standard deviation to assess the estimations of each DNN in Fig. 5c, d. It is seen that the RMSEs of the pre-estimates for DNNs are significantly correlated with

their means and standard deviations. The Pearson correlation coefficients between the means and the RMSEs are −0.9937 in Fig. 5c and −0.9933 in Fig. 5d. Those between the standard deviations and the RMSEs are mostly greater than 0.5 (0.6258 in Fig. 5c and 0.6074 in Fig. 5d). The Pearson correlation coefficients over all cases can be found in Fig. S13. These results reveal that the RMSE of the pre-estimates of each DNN is negatively correlated with the mean and is positively correlated with the standard deviation. In other words,

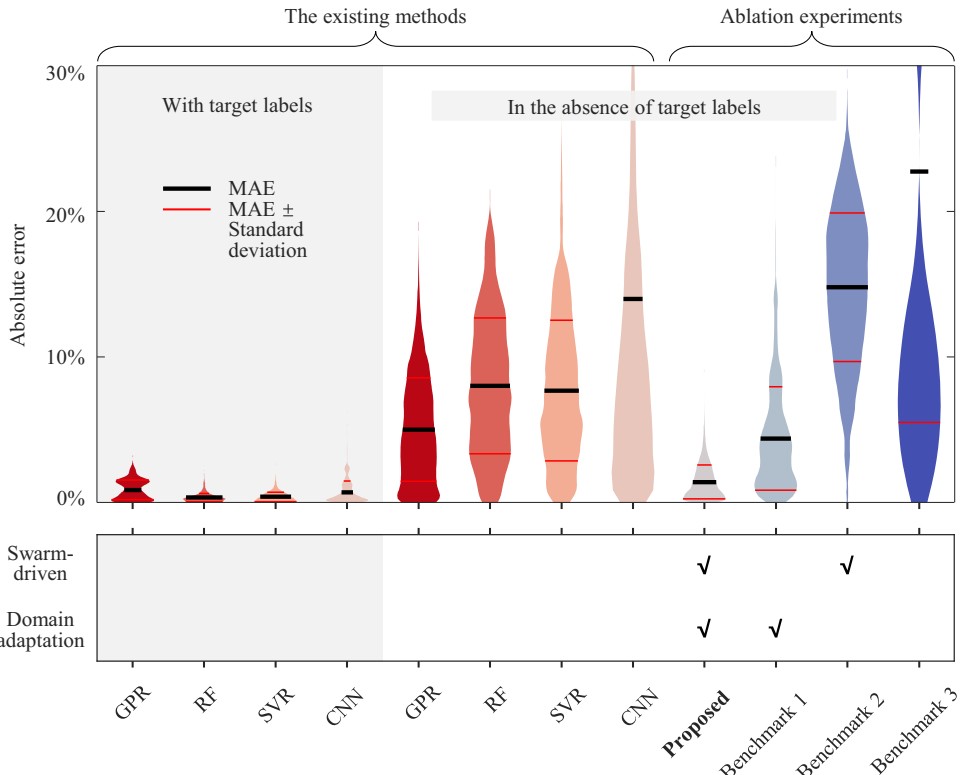

**Fig. 4 | Comparison of absolute error distribution of the SOH estimation.** The shaded region represents the cases with target labels, while the region not shaded represents the cases in the absence of target labels. GPR, RF, SVR, and CNN are representatives of existing methods, which are not equipped with domain adaptation and swarm-driven strategies. Benchmark 1 and Benchmark 2 are designed by disabling the swarm-driven and domain adaptation strategies of the proposed framework, respectively. Benchmark 3 is designed by disabling both strategies.

DNNs whose pre-estimates have a higher mean and lower standard deviation are more likely to have higher accuracy. This explains why excellent DNNs can be selected even in the absence of target labels by only using lower and upper quartiles for the means and standard deviations (theorized in the Methods section). We also report the distributions of pre-estimation RMSEs in Fig. 5a, b corresponding to the selected DNNs (circled in black line) in Fig. 5c, d. It is observed that after the selection, the RMSE bandwidth of the swarm estimation is reduced from over 10% to around 4%, and the mean RMSE decreases to about 2%. These results demonstrate the effectiveness and importance of the DNN swarm-driven strategy in the proposed framework.

After evaluating the swarm-driven strategy, we investigate the other crucial strategy of the proposed framework, i.e., domain adaptation. The feature vectors, before being fed to the terminal fully connected layer, are the subject of domain adaptation. Nevertheless, we do not focus on them but define an explanation map to visualize their contributions to the SOH estimate. The explanation map is defined as the weighted sum of the unflattened weight vector $\mathbf{W}^*$ of the TFC layer and these feature vectors along the channels (theorized in the DNN explanation section). We first dissect an individual DNN without domain adaptation at 10 equal-interval cycles to visualize the evolution of the explanation map as a function of degradation in Fig. 6a. This DNN is first trained for Dataset #1 and then applied to Dataset #5 with no available target labels. It is seen that the explanation maps for Datasets #1 and #5 are dramatically different. When applied to dataset #5 without domain adaptation, the DNN shows abnormally high feature values near the sampling points corresponding to the voltage plateau. As a result, it makes a severe overestimate for Dataset #5. Next, we dissect an individual DNN (with the median RMSE corresponding to Fig. 5a) of the proposed framework to show the explanation maps (see Fig. 6b). Note that a domain-adapted feature vector needs to be

unflattened before being used to compute its explanation map. In contrast to Fig. 6a, the gaps in explanation maps between the two datasets are significantly mitigated by domain adaptation. Thanks to this, the DNN can make accurate estimations even in the absence of the dataset #5 labels. This is the answer regarding the necessity of domain adaptation in the proposed framework.

**Estimation performance with various hyper-parameters**
We further investigate the impact of some crucial hyper-parameters on the estimation performance of the proposed framework, including the size of the DNN swarm, activation functions, number of channels, and number of layers of the CNN. The size of the DNN swarm is set to 1, 50, 100, ..., and 300, respectively, and the results are shown in Fig. 7a. We observe that increasing the swarm size can reduce the overall estimation error. A size of 50 is sufficient for accurate estimation by suppressing the MAE below 2%. Thus, one can balance the accuracy and computational cost by tuning the swarm size in practice. We then examine the influence of activation functions in Fig. 7b by comparing the estimation performance using ReLU, Tanh, Sigmoid, and LogSigmoid, respectively. Note that the Sigmoid before the DNN output is not considered in this comparison as it is used to scale the estimates into [0, 1]. The results show that the ReLU activation function shows the highest accuracy and is therefore preferred when applying the framework. Next, we study the impact of the number of CNN layers and channels on the estimation performance. We span the number of CNN layers from 1 to 4, and the number of channels for all layers is assumed to be identical and belongs to [32, 64, 128, 256]. The results are reported in Fig. 7c. It can be observed that increasing the number of channels does not always reduce the MAE except for the one-layer CNN. The number of channels less than 128 is sufficient to provide an accurate estimation. On the other hand, multiple CNN layers are conducive to high accuracy. One might need to find a suitable number of

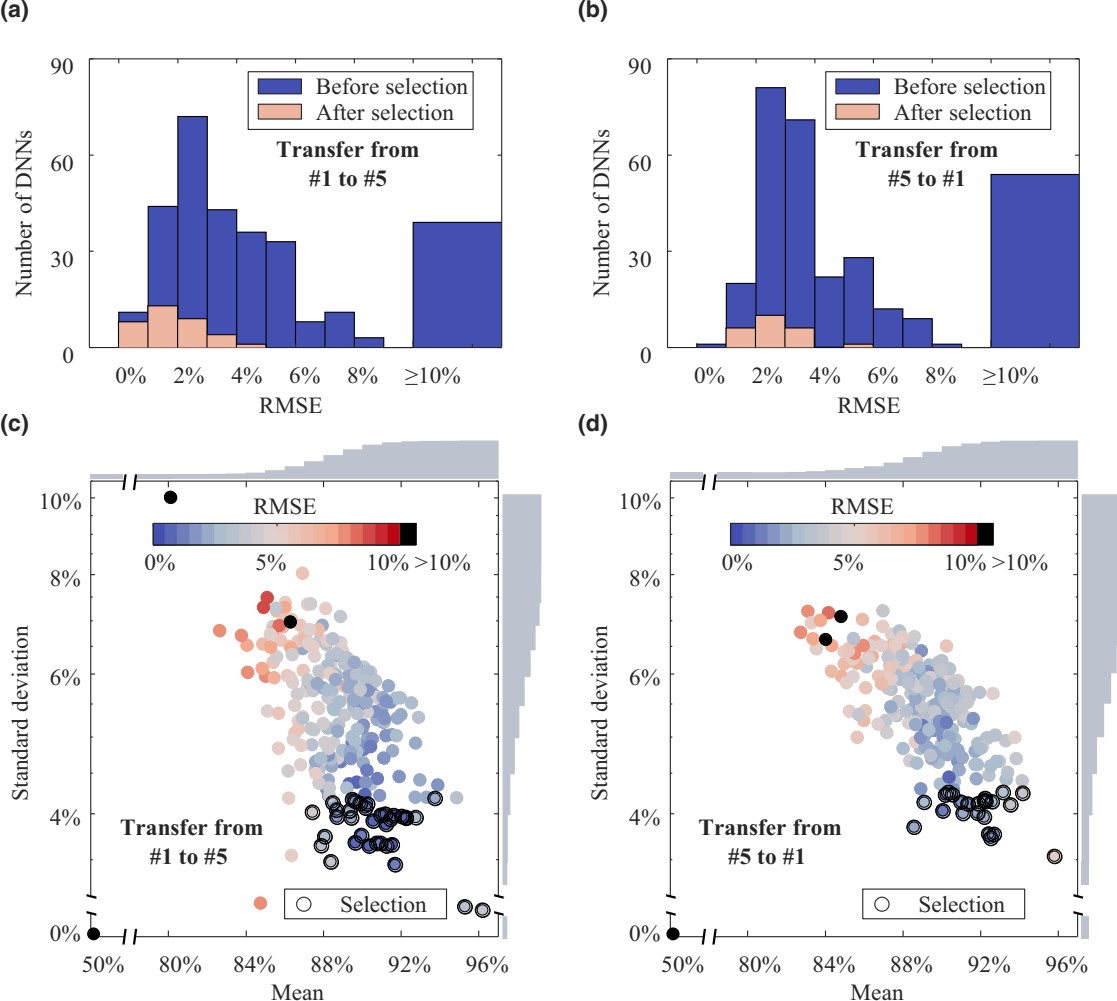

**Fig. 5 | SOH estimation performance of DNNs in the proposed framework.**
**a** Distribution of pre-estimation RMSEs of DNN swarm before and after selection
(transfer from Dataset #1 to #5). **b** Distribution of pre-estimation RMSEs of DNN
swarm before and after selection (transfer from Dataset #5 to #1). **c** Relationship
among the mean, standard deviation, and RMSE of the pre-estimation (transfer
from Dataset #1 to #5). **d** Relationship among the mean, standard deviation, and
RMSE of the pre-estimation (transfer from Dataset #5 to #1). In (**c**, **d**), the histo-
grams along the X and Y axes represent the cumulative distributions of the mean
and standard deviation, respectively.

CNN layers to balance the estimation accuracy and computa-
tional cost.

## Limitations and outlook
The present study can be improved in the future. First, as a data-driven
approach, the proposed framework does not assume specific proper-
ties and dimensions of the input data. Hence, the proposed framework
can be applied to a wider variety of battery materials, other SOH
metrics, and input signals. Second, our preliminary trimming strategy
can be developed with more advanced techniques to optimize esti-
mation performance. Finally, the proposed framework does not
assume specific application scenarios. It thus can be explored to apply
to the big data containing a large amount of battery real-world
operation history. The proposed framework is promising to help
maximize the potential of big data, which generally lacks labels.

## Discussion
Existing techniques for battery SOH estimation are highly dependent
on the labeled degradation data of the target battery, resulting in an
enormous expenditure of time and resources for data collection. In
this work, we devise a target label-agnostic solution to battery SOH
estimation based on deep learning. This framework selectively

integrates the estimations of a swarm of DNNs into a reliable SOH
estimate rather than relying on a single DNN. Each DNN is trained for
source labels and domain invariance of degradation features simulta-
neously. A trim strategy is proposed to regulate the skewness of the
source domain sample distribution to improve the accuracy.

As a case study, we take the partial charging curve as the input of
the proposed framework. For validation, we combine two experi-
mentally generated datasets and three public datasets for cross-vali-
dation, resulting in 80 cases covering 71,588 samples. We first
demonstrate that the proposed framework can achieve absolute errors
of less than 3% for 89.4% of samples (less than 5% for 98.9% of samples),
with a maximum absolute error of less than 8.87% in the absence of
target battery labels. Compared with the existing methods, the pro-
posed framework reduces the MAE and maximum absolute error by
more than half. These results illustrate the successful application for
various domains. Furthermore, we dissect the DNN and visually
explicate that the proposed architecture of DNNs can effectively
minimize the domain gap. The analysis of the swarm of DNNs unveils a
correlation between the mean, standard deviation, and errors of DNNs'
estimates and clarifies how our framework can select the DNNs for
SOH estimation in target label-agnostic cases. Finally, we investigate
the impact of the crucial hyper-parameters on the estimation

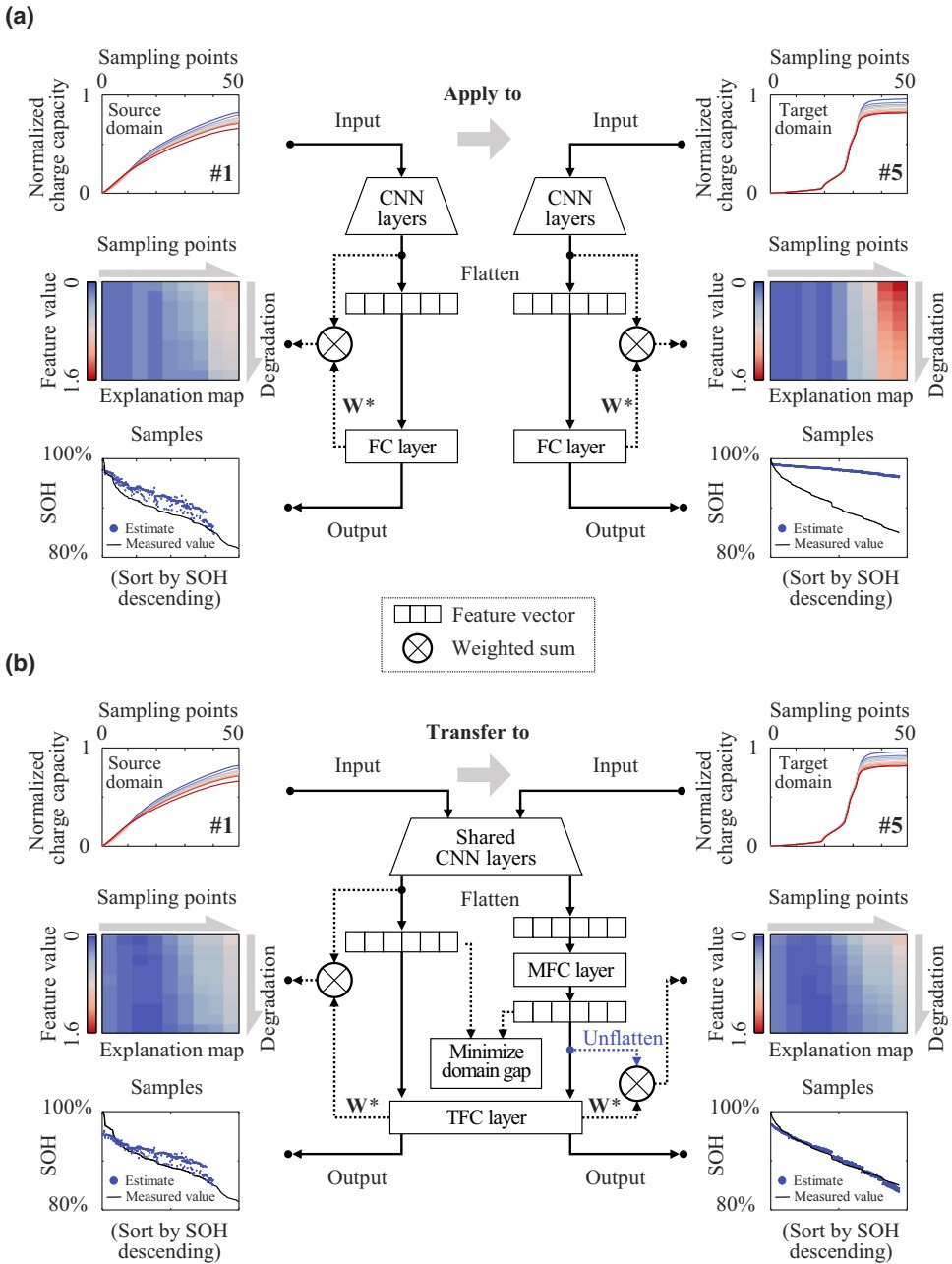

**Fig. 6 | Dissection diagram of the proposed framework (case of transfer from Dataset #1 to #5). a** Diagram of a DNN without domain adaptation. **b** Diagram of the DNN with the median RMSE corresponding to Fig. 5a in the proposed framework.

performance, and provide references for hyper-parameter selection to apply the proposed framework better.

In summary, our work highlights the potential of deep learning in supplanting time- and resource-consuming battery degradation tests. We envisage that the proposed framework will motivate the utilization of large-scale historically collected but unlabeled LIB data (e.g., onboard data, cloud data). It can also enable the rapid development of BMS for new-generation batteries using only existing experimental data.

## Methods
### Data processing

Partial charging curves of LIBs are employed as the input of DNNs for SOH estimation. In a constant-current charging process, the voltage $V(t)$ and current $I(t)$ are stored by BMS at a time step $t$, and the partial charging curve $\mathbf{q}^{\psi}$ can then be captured by setting a voltage sampling window:

$$
\begin{cases}
\mathbf{q}^{\psi} = \left[ \dfrac{Q_0^{\psi}(V)}{Q} \quad \dfrac{Q_1^{\psi}(V)}{Q} \quad \cdots \quad \dfrac{Q_K^{\psi}(V)}{Q} \right], \psi \in \{S, T\} \\
Q_i^{\psi}(V) = \int_{V(t)=V_{\min}}^{V(t)=V_{\min}+i\Delta V} |I(t)| \mathrm{d}t, \ i \in \{0, 1, ..., K\}
\end{cases}
\tag{1}
$$

where the superscript $\psi$ indicates whether $\mathbf{q}^{\psi}$ belongs to the source domain $S$ or the target domain $T$. $Q$ denotes the initial capacity, which is used to normalize the partial charging curve of different types of LIBs. $V_{\min}$ is the lower voltage limit. The voltage sampling window is gridded by a given voltage step $\Delta V$ and ranges from $V_{\min}$ to $V_{\min} + K\Delta V$.

To improve estimation performance, we generate a more balanced source domain by trimming the distribution of samples in the original source domain. Specifically, the original source domain samples are first grouped into $n_{\mathrm{bin}}$ bins with a uniform width (set to 2% in this work) according to their labels. Using the number of samples

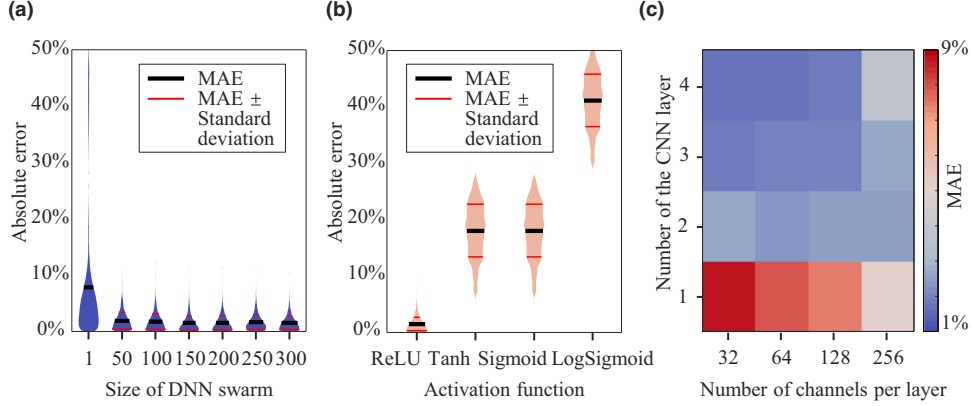

**Fig. 7 | Estimation performance with various hyper-parameters. a** The impact of swarm size. **b** The impact of various activation functions. **c** The impact of the numbers of CNN layers and channels.

$n_{i,\text{origin}}$ in the $i$-th bin as an upper bound, the number of samples in this bin $n_{i,\text{trim}}$ that need a trim is then optimized by:

$$\min_{0 \le n_{i,\text{trim}} \le n_{i,\text{origin}}, n_{i,\text{trim}} \in \not\!\mathbb{c}} \sum_{k=1}^{2} \alpha_k g_k$$
$$\begin{cases} n_{i,\text{remain}} = n_{i,\text{origin}} - n_{i,\text{trim}} \\ g_1 = \left| \mu_3 \left( \bigcup_{i=1}^{n_{\text{bin}}} \{y_{ij}^S\}_{j=1,\dots,n_{i,\text{remain}}} \right) - \mu_3^* \right|, g_2 = \frac{\sum_{i=1}^{n_{\text{bin}}} n_{i,\text{trim}}}{\sum_{i=1}^{n_{\text{bin}}} n_{i,\text{origin}}} \end{cases} \quad (2)$$

where $n_{\text{remain}}$ represents the number of the remaining samples after trim. $\{y_{ij}^S\}_{j=1,\dots,n_{i,\text{remain}}}$ represents $n_{\text{remain}}$ source domain samples randomly selected from the $i$-th bin. $g_k$ denotes the component of the objective function, $k \in \{1,2\}$, and $\alpha_k$ is the weight corresponding to the $g_k$. In this work, $\alpha_1$ and $\alpha_2$ are set to 0.2 and 2, respectively. $\mu_3(\cdot)$ is the skewness operator defined as the third standardized moment:

$$\mu_3(y) = \frac{E(y-\mu)^3}{\sigma^3} \quad (3)$$

where E($\cdot$) is the expectation operator, $\mu$ and $\sigma$ are the mean and standard deviation of $y$, respectively. In this optimization, minimizing $g_1$ makes the skewness of the trimmed source domain approach its target value $\mu_3^*$ (set to zero in this work) to avoid asymmetry, while minimizing $g_2$ ensures that as few samples as possible are discarded. Two constraints on the optimization are defined to generate a new source domain with a comparable range and unimodal distribution:

$$\begin{cases} \frac{\max\limits_{i \in [1, n_{\text{bin}}]} (n_{i,\text{remain}}) - \min\limits_{i \in [1, n_{\text{bin}}]} (n_{i,\text{remain}})}{\sum_{i=1}^{n_{\text{bin}}} n_{i,\text{origin}}} \le \varepsilon \\ \sum_{i=2}^{n_{\text{bin}}-1} \text{sgn}(n_{i+1,\text{remain}} - n_{i,\text{remain}}) - \text{sgn}(n_{i,\text{remain}} - n_{i-1,\text{remain}}) = 1 \end{cases} \quad (4)$$

where $\varepsilon$ is the maximum difference in the number of samples among the bins, which is set to 4.5% in this work.

## DNN architecture

The proposed framework is composed of a swarm of DNNs. For the DNN #$x$, $x \in \{1, 2, \dots, N\}$, the gridded input is first processed by serially stacked 1D CNN layers for feature vector extraction, which can be formulated as:

$$\begin{cases} {}^{l+1}\mathbf{X}_m^\psi(i) = {}^{l+1}\mathbf{b} + \sum_{c=1}^{{}^l C} \sum_{j=1}^{{}^{l+1}k} {}^{l+1}\mathbf{w}_c \otimes {}^l\mathbf{X}_c^\psi \left( {}^{l+1}s_{\text{tr}} i + j \right) \\ m \in \{1, 2, \dots, {}^{l+1}C\}, i \in \{0, 1, \dots, {}^{l+1}L\} \\ {}^{l+1}L = \frac{{}^l L - {}^{l+1}k}{{}^{l+1}s_{\text{tr}}} + 1, l \in \{1, 2, \dots, \Upsilon - 1\} \end{cases} \quad (5)$$

where ${}^{l+1}\mathbf{X}^\psi$ and ${}^l\mathbf{X}^\psi$ denote the output and input of the $(l+1)$-th 1D CNN layer, respectively. $\otimes$ represents the valid cross-correlation operator. ${}^{l+1}\mathbf{w}$, ${}^{l+1}\mathbf{b}$, ${}^{l+1}k$, and ${}^{l+1}s_{\text{tr}}$ are the weight, bias, kernel size, and stride of the $(l+1)$-th 1D CNN layer, respectively. ${}^l C$ and ${}^{l+1}C$ are the numbers of input and output channels of the $(l+1)$-th 1D CNN layer, respectively. ${}^{l+1}L$ and ${}^l L$ are the lengths of the ${}^{l+1}\mathbf{X}^\psi$ and ${}^l\mathbf{X}^\psi$, respectively. $(\Upsilon-1)$ denotes the total number of the 1D CNN layers.

A MFC layer is exclusively designed for the target domain after the shared CNN layers to reconstruct the extracted feature vectors. The output of the terminal 1D CNN layer is flattened by the channel and then input to the MFC layer, which can be described as:

$$^{\text{MFC}}\mathbf{X}^T = \mathbf{W}_{\text{MFC}} \, {}^\Upsilon\mathbf{X}^T + \mathbf{b}_{\text{MFC}} \quad (6)$$

where ${}^\Upsilon\mathbf{X}^T$ and ${}^{\text{MFC}}\mathbf{X}^T$ are the target domain input and output of the MFC layer, respectively. $\mathbf{W}_{\text{MFC}}$ and $\mathbf{b}_{\text{MFC}}$ are the weight and bias vectors of the MFC layer, respectively.

A shared TFC layer is designed at the terminal of the DNN for both source and target domains to regress SOH. The source domain output of the terminal 1D CNN layer is flattened and then fed to the TFC layer, while the target domain output of the MFC layer is directly provided to the TFC layer. This layer can be expressed as:

$$\vartheta^\Psi = \begin{cases} \mathbf{W}_{\text{TFC}} \, {}^\Upsilon\mathbf{X}^S + b_{\text{TFC}}, & \text{if } \Psi = S \\ \mathbf{W}_{\text{TFC}} \, {}^{\text{MFC}}\mathbf{X}^T + b_{\text{TFC}}, & \text{otherwise} \end{cases} \quad (7)$$

where $\vartheta^\psi$ denotes the SOH pre-estimate of each DNN. ${}^\Upsilon\mathbf{X}^S$ is the source domain output of the $\Upsilon$-th 1D CNN layer. $\mathbf{W}_{\text{TFC}}$ and $b_{\text{TFC}}$ are the weight vector and the bias of the TFC layer, respectively.

The rectified linear unit (ReLU) activation function, which takes the maximum value between 0 and its input as output, is designed to follow each 1D CNN layer and fully connected layer. The sigmoid activation function is applied before outputting the estimate to ensure that the SOH pre-estimate is between 0 and 1.

## DNN explanation

We define a vector $\mathbf{F}^{\psi}$ to visualize the domain adaptation of the proposed framework, which can be formulated as:

$$\mathbf{F}^{\psi} = \sum_{1}^{\zeta C} \mathbf{W}^{*} \odot {}^{\zeta}\mathbf{X}^{\psi}, \psi \in \{S,T\}, \zeta \in \{\Upsilon, \mathrm{TFC}\} \qquad (8)$$

where $\odot$ denotes the element-wise multiplication of two vectors. $\mathbf{W}^{*}$ is the unflattened weight vector of the TFC layer. ${}^{\zeta}\mathbf{X}^{\psi}$ is the unflattened feature vector or the feature vector before flattening. The sigmoid activation function is also applied before the output of this equation. Thus, $\mathbf{F}^{\psi}$ is equivalent to a TFC layer output minus $b_{\mathrm{TFC}}$ and cancels the sampling point-wise weighted sum. Combining $\mathbf{F}^{\psi}$ at various SOH can produce an explanation map for visualizing the contribution of the feature vectors to the lifelong SOH estimation.

## DNN training

First, $N$ DNNs are independently trained, and in the present study, we set $N = 300$. Each DNN is parameterized by the labeled data from the source domain and the unlabeled data from the target domain. The widely-used Adam algorithm[48] is employed to optimize the parameters iteratively. The learning rate is set to 0.001. To realize cross-domain transfer learning, we define a loss function $E$ containing three components, which can be formulated as:

$$
\begin{cases}
J = \sum\limits_{i=1}^{3} \kappa_i f_i \\
f_1 = \frac{1}{n_{\mathrm{s}}} \sum\limits_{i=1}^{n_{\mathrm{s}}} (\vartheta_i^S - {}^{*}\vartheta_i^S)^2 \\
f_2 = \left\| \frac{1}{n_{\mathrm{s}}} \sum\limits_{i=1}^{n_{\mathrm{s}}} \phi({}^{\Upsilon}X_i^S) - \frac{1}{n_{\mathrm{t}}} \sum\limits_{i=1}^{n_{\mathrm{t}}} \phi({}^{\mathrm{MFC}}X_i^T) \right\|_H \\
f_3 = \frac{1}{Z} \sum\limits_{i=1}^{Z} (\vartheta_{0,i}^T - 1)^2
\end{cases}
\qquad (9)
$$

where $f_i$ denotes the component of the loss function $J$, $i \in \{1, 2, 3\}$, and $\kappa_i$ is the weight corresponding to the $f_i$. This work sets $\kappa_1$, $\kappa_2$, and $\kappa_3$ to 1, 0.1, and 1, respectively. ${}^{*}\vartheta^S$ denotes the available label of the source domain sample. $n_{\mathrm{s}}$ and $n_{\mathrm{t}}$ are the number of samples from the source and target domains, respectively. $\|\cdot\|_H$ represents the norm of the reproducing Hilbert space in terms of the embedding kernel $\phi(\cdot)$, and the Gaussian kernel is employed as the $\phi(\cdot)$. $\vartheta_0^T$ denotes the pre-estimate of the target domain at the first cycle, and $Z$ is the number of these pre-estimates. $f_1$ evaluates the mean squared error between each element of the source domain in the pre-estimate and the available labels, which is designed for training each DNN to learn SOH estimation from labeled source domain samples. $f_2$ evaluates the domain invariance, and the maximum mean discrepancy (MMD) is employed as a criterion to measure the distance between the high-dimensional degradation features of the source domain samples and those of the target domain samples after reconstruction. $f_3$ is also the measure of the mean squared error but only for the target domain samples at the first cycle. This is because the partial charging curves of the first cycle (i.e., in fresh status) are easily obtained (e.g., by LIB formation or factory test), and their labels can be treated as 1 to improve the learning of the target domain samples.

In each sub-training, samples from the source domain are divided into a training set and a validation set. Two-thirds of the source domain samples are used as the training set, and the rest are the validation set. Each sub-training is terminated when the RMSE of the validation and training sets is less than both 5%, or when the number of epochs reaches 2000. The minimum number of epochs is set to 500. All the samples are divided into mini-batches for training with a mini-batch size of 20. All DNNs are trained based on an NVIDIA Tesla V100 GPU in this work.

## SOH estimation with trained DNNs

The proposed framework selectively integrates the pre-estimates of the DNNs to generate a reliable SOH estimate. We employ mean and standard deviation to evaluate the pre-estimates of each trained DNN. An efficient metric, quartile, is used to select DNNs according to these measures. The retaining DNNs are the final choices of the proposed framework for each estimate, and the indexes $\mathbf{x}$ of the final choices can be formulated as:

$$\mathbf{x} = \{x | \mathrm{E}({}_x\vartheta^T) \geq Q_3(\mathrm{E}({}_x\vartheta^T)), \mathrm{Var}({}_x\vartheta^T) \leq Q_1(\mathrm{Var}({}_x\vartheta^T))\} \qquad (10)$$

where $Q_1$ and $Q_3$ denote the lower quartile operator and the upper quartile operator, respectively. The expected pre-estimates of the retaining DNNs can be treated as the final SOH estimation for the target domain. In this study, we employ root mean square error RMSE, absolute error AE, and its average value MAE to evaluate the SOH estimation:

$$
\begin{cases}
\mathrm{RMSE} = \sqrt{\frac{\sum_{i=1}^{M}(y_i - \hat{y}_i)^2}{M}} \\
\mathrm{AE}_i = |y_i - \hat{y}_i| \\
\mathrm{MAE} = \frac{\sum_{i=1}^{M} \mathrm{AE}_i}{M}
\end{cases}
\qquad (11)
$$

where $y_i$ and $\hat{y}_i$ are the measured value and the final estimate of the SOH for sample $i$, respectively. $M$ represents the total number of samples of interest.

## Battery cycling and dataset generation

Batteries from datasets #3 and #5 are tested in thermal chambers at 20 and 45 °C, respectively. An ARBIN BT2000 battery test system is employed to cycle the batteries. In each cycle of the battery from Dataset #3, the charge strategy is to charge at a constant-current rate of 0.3 C until the voltage reaches 4.2 V and then hold at 4.2 V until the charging current drops below 0.03 A; the discharge strategy is to discharge at a constant-current rate of 2 C. In each cycle of the battery from Dataset #5, the charge strategy is to charge at a constant-current rate of 1 C until the voltage reaches 3.65 V and then hold at 3.65 V until the charging current drops below 1.35 A; the discharge strategy is to discharge at a constant-current rate of 1 C. The charging curves extracted from the constant-current charging phase of the cycles are integrated into the datasets.

## Data availability

Datasets #3 and #5 generated in this study have been deposited in the Mendeley database under the accession code: https://data.mendeley.com/datasets/v8k6bsr6tf/1.

## Code availability

Code for the modeling work is available from the corresponding authors upon request.

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

## Acknowledgements

This work was funded by the National Key R&D Program of China under Grant 2021YFB2402002 (R.X.), the Beijing Natural Science Foundation under Grant L223013 (R.X.), and the China Postdoctoral Science Foundation under Grant BX2021035 and 2022M710379 (J.T.).

## Author contributions

R.X. conceived the idea of SOH estimation, led and supervised the project, participated in paper writing and revision, and provided guidance to all co-authors. F.S. supervised and led this project. J.L., J.T., and C.W. generated the data. J.L. conceived, wrote, and revised the manuscript. All the authors have revised the manuscript and agreed with its content.

## Competing interests

The authors declare no competing interests.
