## [Peer review file · Nature Communications]

REVIEWER COMMENTS

Reviewer #1 (Remarks to the Author):

In this work, the authors proposed a deep transfer-learning framework for SOH estimation in the absence of labeled target data. The proposed framework trains a swarm of DNNs to accurately estimate battery cell SOH regardless of the cell chemistry and in the absence of labeled target data. The reproducibility of the framework is demonstrated through multiple independent runs with various hyperparameters. The methodology is well thought out, and the method's applicability to different battery chemistries is impressive. Additionally, the figures are well made and tell a good story. However, the paper could be further improved. The comments below should help the authors improve their paper. The comments below are not exhaustive, and overall, the writing in this paper needs significant elaboration, revision, and polish.

Comments on the content

1. Overall Contribution – The authors used a deep learning-enabled framework in which they trained the model without needing additional target-labeled data to estimate the SOH. Also, the considered framework is domain adaptation-enabled which can be employed for cross-manufacturer battery SOH estimation. This work can motivate the use of large-scale field data without SOH labels and can be a substitute for degradation experiments of the target domain.
2. Overall Methodology – The authors developed a SOH estimation framework composed of a swarm of DNNs and driven by a dropout rule inspired by the Monte-Carlo sampling. Each DNN is trained for source labels and domain invariance of degradation features simultaneously and then selected by a round of self-checking. For the case study, they take the partial charging curve as the input of the framework from a combined dataset covering 80 cases and 71588 cycles to determine the prediction accuracy. One suggestion the reviewer has for the authors is to benchmark the results for the case study from conventional methods and the proposed framework in terms of errors, and computational and experimental costs, so that the audience could better understand the improvements this work can offer. It would be better if the authors could show the trade-off between the complexity of the framework and the costs of experimental procedures.
3. Introduction – The authors state: “In general, SOH is defined as the ratio between the present capacity to the initial maximum capacity 9,10.” However, using both the words “initial” and “maximum” is misleading because some cells' initial capacity may be less than their maximum capacity. Some cells can gain capacity in the first few days of operation, as seen in the dataset published by Severson et al. [1]. The authors are suggested to remove the word “maximum” to clarify the statement.
4. Introduction – The authors did a fair job describing existing approaches to deep learning domain adaptation problems. However, the connection between the cited works and the author's proposed method should be further elaborated on. It is unclear how the proposed method differs from existing methods in the literature. What new ideas and methods do the authors propose that have not yet been

investigated? In particular, the reviewers noticed few citations in the final paragraph of the Introduction. It would strengthen the rigor of the paper if the authors improved the connection between their proposed method and existing research by citing and elaborating on it further.

5. Results – Data Generation – The authors say, “Without losing generality, we adopt the partial charging data as the input of the model for SOH estimation, as has been demonstrated by previous studies^{23,33–36}. A 500-mV voltage window is utilized to extract the partial charging data and it covers 3.5 V to 4 V for Cell #1~4 and 3.1 V to 3.6 V for Cell #5” The authors cite other research papers which use partial charging data for SoH estimation, but do not elaborate on the rationale behind this decision? Why choose to use partial charging curves? Please provide more insight. Additionally, what impact does the selected voltage range have on the proposed algorithm?

6. Results – Data Generation – The authors say, “The employed five types of cells are pairwise combined for cross-validation. This results in a total of 20 combinations, each of which consists of four cases with 4 different lower SOH bounds.” This sentence is confusing for readers and does not fully explain how the cross validation is performed. Please provide more details on the cross-validation scheme.

7. Results – Comparison with existing methods – The authors title this section “Comparison with existing methods,” but do not mention what the existing methods are. There is no background explanation on why Benchmark 1 and Benchmark 2 are good comparisons for the proposed method. The authors are advised to better describe the methods being compared. Furthermore, what do the authors mean that benchmark II treats the cycle labels as “100%.” What does 100% mean? This is unclear. Here is the full sentence for reference, “Benchmark II is generated based on Benchmark I, and additionally treats the initial cycle labels of the target domain as 100% and includes them in training.” The authors are advised to clarify the meaning of this sentence.

8. Results – Comparison with existing methods – In Figure 7, the authors show a very limited number of capacity predictions. From the figure, it is hard to get a sense of what “3%” error looks like on a real cell. The authors are advised to revise Figure 7 to include more prediction points or are suggested to create a new figure showing predicted values overlaid on a capacity-fade curve.

9. Results – Cross-manufacturer battery SOH estimation in the absence of target labels – The authors highlighted the importance of domain balance in estimating without target labels but didn’t clarify the effects of domain imbalance and trimming process in estimation performance. The reviewers suggest that the authors clarify the potential effects of skewed SOH distribution on the estimation performance and incorporate the techniques for handling that in the framework.

10. Conclusion – The authors conclude that “proposed framework is friendly for further applications.” This sentence is not clear and could be confusing for the readers. Please provide more details to clarify how multiple runs with various hyperparameters show the friendliness of the framework for further applications.

1. The authors repeatedly misuse the word “herein” and other common conjoining words (nevertheless, however, etc.) in the paper. The authors are advised to avoid using these words unless necessary and to proofread their paper using advanced grammar-checking software (such as Grammarly) to further improve their paper’s sentence structure and overall clarity.
2. The authors are misusing the word “rational” and its other forms (e.g., “rationalize”). Please revise sentences using this word. Consider replacing it and only using when absolutely necessary.
3. Please check the grammar structure of the sentences for better readability in:
 - a. The Conclusion section, line 6
 - b. The Conclusion section, line 22
 - c. The Framework Overview section, line 17

References

[1] K. A. Severson et al., “Data-driven prediction of battery cycle life before capacity degradation,” *Nat. Energy*, vol. 4, no. 5, pp. 383–391, 2019, doi: 10.1038/s41560-019-0356-8.

Reviewer #2 (Remarks to the Author):

The manuscript tackles the lack of SOH labels for SOH estimation by a framework aggregating several deep learning networks.

The approach seems convincing from a theoretical point of view. My major concern is that the hyperparameter tuning of the DNNs is insufficient (not well documented & only few trials). Also, the swarm approach is interesting, but a benchmark to a single DNN with a similar number of parameters like the whole DNN swarm is missing. In the section “Comparison with existing methods” no existing methods are used for comparison, only benchmarks defined by the authors are defined. This hinders comparability to state of the art work. It would be interesting to compare models with and without the absence of the target labels (i.e. how much worse does the SOH estimation get with absence of the target labels compared to available target labels? How much better is the SOH estimation with absence of the target labels and your approach?).

It would be handy to have linked cross-reference in the document to Figures, Sections, etc. Some sentences are quite long & nested and, thus, hard to understand. Figures should be vectorized (e.g. Fig 3 is not).

Abstract:

- „the time-consuming experiments” are mentioned twice but have never been mentioned before. In what respect and why are they time-consuming? Are they maybe also expensive?
- “deep learning networks” is not a common term. Do you mean deep neural networks or other deep learning models? Please be more specific.
- Does “absolute error” refer to MAE?
- “within 3% for up to 88.6% of the cycles”: in the Intro this is formulate more precisely and understandably (“framework can achieve absolute errors of 3% for 88.6% of samples”). It seems that the results are little cherry-picked. At least you should give information about the remaining 11.4% of the samples. How bad is the prediction for those? Why is the prediction performance of those samples not relevant?

Intro:

- “In general, SOH is defined as the ratio between the present capacity to the initial maximum capacity”: In general the SOH is not defined capacity-based, but in most cases! There are other ways to define the SOH, e.g., referring to internal resistance, number of charge/discharge cycles that you only mention later in “Limitations and outlook”. The intro is a good spot to already do that and argue, why you opt for the capacity-based SOH.
- “Standard SOH calibration requires fully charging or discharging a battery between the voltage limits”: this is not sufficient. Also, temperature and current profile have an influence. Simply charging or discharging until the voltage limits is not sufficient.
- “A growing number of literatures”: “literature” is a singular-only noun

Framework overview:

- “driven by a dropout rule”: without a look at the figure it is unclear that the dropout rule does not refer to single neurons, but to complete DNNs.

Why did you opt against a single DNN with neuron-wise dropout during training?

- “The current is scaled to C-rate to cope with inconsistent operating current range among various types of LIBs.”: Using the C-rate makes sense. But what do you exactly mean by “inconsistent operating current range”? Please provide an example.

- “Each DNN in the proposed framework is structurally identical (Fig. 1b) but independently trained with different initialized parameters”: Do you mean the hyperparameters are identical (no. layers, no. neurons), but the weights and biases are initialized differently? How are they initialized: Same method (e.g. He, Glorot), but different random seed?
- “respectively” mean “in the order mentioned” – but the order of the domains/branches does not in this case.
- “terminal fully connection layer (TFC)” should be “terminal fully connection (TFC) layer” because “layer” is not part of the acronym. Same for MFC
- “firstly” -> “first”

Data Generation

- In Table 1: “2.7~4.2”: Tilde should be „-“
- Why are you only considered each one cell from the five data sets? I assume even within each of the data sets there is intrinsic cell-to-cell variability from production, but also due to different operational protocols. How could this be considered?
- Fig 2. “SOH distributions”: I have never seen this type of SOH-residence-based visualization for SOH trajectories. I see why you opt for this type of visualization in contrast to the common “cycle vs. SOH” plots given your further plots, e.g. in Fig. 3 c) & d). I would appreciate a little explanation of how to interpret this plot (e.g. sth. like binning/ buckets of SOH values)

Cross manufacturer battery SOH estimation ...

- Fig. 3 a): y-axis label “Error” is not precise – which error do you use exactly? RMSE, MSE, MAE, MAPE,...? This does also apply to the other figures.
- “within 4.90% for the lower SOH bounds” & “within 5.01% even when the domains are extremely imbalanced (i.e. 95% lower SOH bound)”: Where exactly is that visible in Fig. 3? I see errors up to 10 %, e.g. source domain #4 & target domain # 1.
- “In other words, cases with a lower SOH bound of 95%”: Isn’t 95% a higher bound than e.g. 80%? (95>80%)? This is already confusing in previous sentences.
- How do you trim the dataset of cell #2? This is a very interesting point because battery datasets will mostly be imbalanced so dealing with that is also relevant for other models & approaches. How do you make sure you don’t cherry-pick?
- “up to 88.6% cycles with absolute error less than 3%”: This is very confusing wording to me because it may suggest you are having “cycles” as an output. What does “88.6% cycles” mean? I interpret it refers to 88.6% of the cycles in the dataset, having one sample per cycle.

Comparison with existing methods

- The title of this section is misleading because no “existing methods” are cited or mentioned in this section. Only benchmarks proposed by the authors of this work are evaluated. This limits comparability to other work.
- “Additionally treats the initial cycle labels of the target domain as 100%”: As 100% of what? Are you assuming 100% SOH for all initial cycles in the target domain?
- “Thus in the label-agnostic”: Comma missing after “thus”

Performance over multiple independent runs with various hyper-parameters

- “no feedback from the target domain is available for refining these hyper-parameters.”: That is indeed true. Potentially, if you had feedback, e.g. by assuming SOH = 100 & at the first cycle as you do in your benchmarks: How would you change the number of layers, number of neurons, kernel size, etc. while doing the transfer learning? I am not aware of any approaches accomplishing that. Thus, I am curious how you would accomplish that. If you are not considering that I suggest a rephrasing.
- “Herein we do not discuss”: Comma missing after “herein”
- “Instead, we conduct multiple independent runs with various hyperparameters to verify the sensitivity of the proposed framework to the hyper-parameter settings”: Your hyperparameter tuning with “additional six groups of randomly formulated hyperparameters” does not convince me regarding optimality of the hyperparameter choice. What is the range of number of layer, number of neurons, activation function, regularization/dropout/early stopping you have considered? Furthermore, six random trials seems very little to me. I would prefer using some established Bayesian optimization approach for hyperparameter tuning. Or also more random trials.

Appendix/ Methods:

- Putting the equations of Adam is not necessary or interesting for any reader in my opinion. Adam is a well-established, standard algorithm so referring to the initial, proposing paper is sufficient.

Responding letter to the reviewers' comments

Journal Title: *Nature Communications*

Manuscript Title: *Deep learning to estimate battery state of health without additional degradation experiments*

Note of thanks to reviewers:

The authors would like to express their appreciation to the reviewers for their time and comments. The manuscript has been systematically revised in its technical and literal content compared to its earlier version. Point-by-point clarifications to the reviewers' comments are listed below, followed by the modifications introduced in the document where appropriate. Our responses are in blue, while our changes to the manuscript are in red.

Comments from the reviewers:

Reviewer #1:

General comment: In this work, the authors proposed a deep transfer-learning framework for SOH estimation in the absence of labeled target data. The proposed framework trains a swarm of DNNs to accurately estimate battery cell SOH regardless of the cell chemistry and in the absence of labeled target data. The reproducibility of the framework is demonstrated through multiple independent runs with various hyperparameters. The methodology is well thought out, and the method's applicability to different battery chemistries is impressive. Additionally, the figures are well made and tell a good story. However, the paper could be further improved. The comments below should help the authors improve their paper. The comments below are not exhaustive, and overall, the writing in this paper needs significant elaboration, revision, and polish.

Thank you for your recognition of our focus and positive comments. Estimating battery SOH without additional degradation experiments is highly challenging but very important for further development. It is promising to help the rapid development of battery management systems for new-generation batteries using only existing experimental data, saving

considerable time and resources. It is also expected to motivate the utilization of large-scale field data without labels.

We also appreciate your comments pointing out the shortcomings of our original manuscript. We have carefully considered all the comments and suggestions. The revised manuscript has been significantly elaborated, revised, and polished. The following are detailed responses to all the comments.

Comment 1: Overall Contribution – The authors used a deep learning-enabled framework in which they trained the model without needing additional target-labeled data to estimate the SOH. Also, the considered framework is domain adaptation-enabled which can be employed for cross-manufacturer battery SOH estimation. This work can motivate the use of large-scale field data without SOH labels and can be a substitute for degradation experiments of the target domain.

Thank you again for your recognition of our contributions. Experiments to collect lifelong battery data can last months or years and should cover considerable resources & many different conditions, yet are essential to existing SOH estimation methods. Thus, this is undoubtedly a primary bottleneck for the further development of batteries. By integrating a swarm of domain adaptation-enabled deep neural networks, our method can make accurate SOH estimation for a given battery without performing degradation experiments, saving a lot of time and resources.

Comment 2: Overall Methodology – The authors developed a SOH estimation framework composed of a swarm of DNNs and driven by a dropout rule inspired by the Monte-Carlo sampling. Each DNN is trained for source labels and domain invariance of degradation features simultaneously and then selected by a round of self-checking. For the case study, they take the partial charging curve as the input of the framework from a combined dataset covering 80 cases and 71588 cycles to determine the prediction accuracy. One suggestion the reviewer has for the authors is to benchmark the results for the case study from conventional

methods and the proposed framework in terms of errors, and computational and experimental costs, so that the audience could better understand the improvements this work can offer. It would be better if the authors could show the trade-off between the complexity of the framework and the costs of experimental procedures.

We greatly appreciate your insightful suggestions.

In the revised manuscript, we have compared the proposed framework with conventional methods in terms of error, experimental cost, and computational cost in the “*Comparison with existing methods*” section (Page 15). Specifically, we have made the following changes.

First, we have made a comparison in terms of the error in the “*Comparison with existing methods*” section (Page 15). The comparative results of the absolute error distribution are described in Fig. 4, and detailed results are shown in Supplementary Note 8. Four popular methods including Gaussian process regression (GPR), random forest (RF), support vector regression (SVR), and convolutional neural network (CNN) are employed to represent conventional methods. The results show that these methods achieve high accuracy when the target labels are available, but fail to provide reliable estimation without target labels. By contrast, the proposed framework achieves accurate SOH estimation without target labels, reducing the MAE and maximum absolute error by more than half.

Second, we have made a comparison in terms of the experimental cost. We estimate the experimental cost of collecting the target labels needed for the conventional methods. The estimation process can be found in Supplementary Note 2, and the estimated results are listed in Table 1 (Page 9). The results show that developing the target labels requires 644-8473 hours of degradation experiments. The high accuracy of these conventional methods comes at the cost of numerous workforce and energy. By comparison, the proposed framework makes accurate SOH estimation without target labels, thereby significantly alleviating the experimental cost.

Third, we have made a comparison in terms of the computational cost in Supplementary Note 9. The results show that the conventional methods have advantages in computational cost due to their simple structures, but their accuracy is unacceptable in the absence of target labels. By contrast, the proposed framework takes merely about 0.7 hours to train 300 DNNs, which avoids conducting 644-8473 hours of degradation experiments. More importantly, we

demonstrate in the “*Estimation performance with various hyper-parameters*” section (Page 21) that a swarm size of 50 is sufficient for accurate estimation. Thus, one can balance the accuracy and computational cost by tuning the swarm size in practice.

Comment 3: Introduction – The authors state: “In general, SOH is defined as the ratio between the present capacity to the initial maximum capacity^{9,10}.” However, using both the words “initial” and “maximum” is misleading because some cells’ initial capacity may be less than their maximum capacity. Some cells can gain capacity in the first few days of operation, as seen in the dataset published by Severson et al. [1]. The authors are suggested to remove the word “maximum” to clarify the statement.

Thank you so much for your professional suggestion.

In the revised manuscript, we have removed the word “maximum” to clarify the statement (Introduction section, Page 3, Line 14): “... SOH defined as the ratio between the present capacity and the initial capacity is drawing broad attention...”.

Comment 4: Introduction – The authors did a fair job describing existing approaches to deep learning domain adaptation problems. However, the connection between the cited works and the author’s proposed method should be further elaborated on. It is unclear how the proposed method differs from existing methods in the literature. What new ideas and methods do the authors propose that have not yet been investigated? In particular, the reviewers noticed few citations in the final paragraph of the Introduction. It would strengthen the rigor of the paper if the authors improved the connection between their proposed method and existing research by citing and elaborating on it further.

We greatly appreciate your rigorous and professional comment.

To strengthen the rigor of the manuscript, we have carefully revised the “Introduction” section (Pages 3-5). We first explain in the second paragraph that SOH cannot be directly measured but requires estimation. Next, in the third and fourth paragraphs, we mainly report that the existing methods devote to extracting features correlated with SOH degradation and

mapping them to the SOH, but are very limited by their dependency on target labels. Many studies^{21,22,25,26,27} on estimating SOH with deep neural networks (DNNs) focus on enhancing the ability of automatic degradation feature extraction, but they still cannot get rid of this dependence. In the fifth paragraph, we show that DNNs can accomplish tasks without target labels in the field of visual recognition³³. However, LIB SOH estimation, which has a massive demand for target labels, has yet to benefit from it. To this end, we propose a novel deep learning-based framework to estimate battery SOH without relying on target labels. Finally, we point out the innovations of the proposed method over existing methods in the last paragraph. Our method “integrates the estimates of a swarm of DNNs into a reliable SOH estimate rather than relying on a single DNN”. Also, “individual DNNs are trained to learn cross-domain knowledge according to source labels and domain invariance of degradation features”. More importantly, we “reveal the influence of the sample distribution in the source domain on SOH estimation and propose to improve estimation performance by trimming the sample distribution in the source domain”. On this basis, the connection between the existing works and the proposed method can be more clearly articulated.

The reference to this response is as follows:

21. Yang, N., Song, Z., Hofmann, H. & Sun, J. Robust State of Health estimation of lithium-ion batteries using convolutional neural network and random forest. *J Energy Storage* 48, (2022).
22. Li, P. et al. State-of-health estimation and remaining useful life prediction for the lithium-ion battery based on a variant long short term memory neural network. *J Power Sources* 459, 228069 (2020).
25. Tian, J., Xiong, R., Shen, W., Lu, J. & Yang, X. G. Deep neural network battery charging curve prediction using 30 points collected in 10 min. *Joule* 5, (2021).
26. Shu, X. et al. A Flexible State-of-Health Prediction Scheme for Lithium-Ion Battery Packs with Long Short-Term Memory Network and Transfer Learning. *IEEE Transactions on Transportation Electrification* 7, (2021).
27. Tan, Y. & Zhao, G. Transfer Learning With Long Short-Term Memory Network for State-of-Health Prediction of Lithium-Ion Batteries. *IEEE Transactions on Industrial Electronics* 67, 8723–8731 (2020).
33. MacDonald, T., Gibson, C. T., Constantopoulos, K., Shapter, J. G. & Ellis, A. v. Deep Domain Confusion: Maximizing for Domain Invariance *Eric. Appl Surf Sci* 258, (2012).

Comment 5: Results – Data Generation – The authors say, “Without losing generality, we adopt the partial charging data as the input of the model for SOH estimation, as has been

demonstrated by previous studies^{23,33–36}. A 500-mV voltage window is utilized to extract the partial charging data and it covers 3.5 V to 4 V for Cell #1~4 and 3.1 V to 3.6 V for Cell #5” The authors cite other research papers which use partial charging data for SoH estimation, but do not elaborate on the rationale behind this decision? Why choose to use partial charging curves? Please provide more insight. Additionally, what impact does the selected voltage range have on the proposed algorithm?

We greatly appreciate your professional comment.

To clarify this, we have made the following changes in the revised manuscript.

First, we have added a description (*“Framework overview” section, Page 6, Line 8*) to clarify the reasons for using the partial charging curves: *“...Without loss of generality, the battery charging data are employed as the input of the DNNs since the battery charging process is generally controllable and occurs regularly...”*. Thus, the battery charging data are employed as the input of our method without losing generality.

Second, following your insightful suggestions, we have investigated the impact of different voltage ranges on SOH estimation performance in Supplementary Note 5. The impact is investigated by varying the window length from 300 to 800 mV at a step of 50 mV. The results show that the proposed framework achieves accurate estimation with an absolute error below 2.4% in most cases. Thus, one might need to determine an appropriate window length to balance estimation accuracy and sampling duration according to application scenarios. We have added an index to this investigation at the end of the first paragraph in the *“Cross-dataset battery SOH estimation in the absence of target labels”* section: *“... The impact of different voltage windows on SOH is investigated in Supplementary Note 5.”* (Page 11, Line 15)

Comment 6: Results – Data Generation – The authors say, “The employed five types of cells are pairwise combined for cross-validation. This results in a total of 20 combinations, each of which consists of four cases with 4 different lower SOH bounds.” This sentence is confusing for readers and does not fully explain how the cross validation is performed. Please provide more details on the cross-validation scheme.

We greatly appreciate your rigorous comment.

To clarify this, we have made the following changes.

First, we have revised the first paragraph of the “*Cross-dataset battery SOH estimation in the absence of target labels*” section (Page 11) to clarify the cross-validation scheme. The cross-validation scheme is designed to evaluate the performance of cross-dataset battery SOH estimation. The employed five types of LIBs are pairwise combined for cross-validation, resulting in a total of 20 combinations. Furthermore, we postulate that the SOH of batteries in the target domain are distributed from 100% to 95%, 90%, 85%, and 80%, respectively. Thus, each combination contains 4 cases with 4 different lower SOH bounds.

Second, we have also added a table to provide more details on the cross-validation scheme in Supplementary Note 4. The cross-validation scheme can be better understood based on this table.

Comment 7: Results – Comparison with existing methods – The authors title this section “Comparison with existing methods,” but do not mention what the existing methods are. There is no background explanation on why Benchmark 1 and Benchmark 2 are good comparisons for the proposed method. The authors are advised to better describe the methods being compared. Furthermore, what do the authors mean that benchmark II treats the cycle labels as “100%.” What does 100% mean? This is unclear. Here is the full sentence for reference, “Benchmark II is generated based on Benchmark I, and additionally treats the initial cycle labels of the target domain as 100% and includes them in training.” The authors are advised to clarify the meaning of this sentence.

We greatly appreciate your professional and valuable comments.

In this work, SOH is defined as the ratio between the present capacity and the initial capacity. On this basis, the label of the initial cycle for any battery is 100%. To avoid confusion, we have removed this statement from the manuscript. To make a clearer comparison, we have revised the “*Comparison with existing methods*” section (Page 15). Compared with the previous version, the revised version mainly focuses on the following changes.

First, we have clarified the background of the methods involved in the comparison. Four popular methods from the literature are employed as representatives of existing methods, including Gaussian process regression³⁵ (GPR), random forest⁴⁴ (RF), support vector regression⁴⁵ (SVR), and convolutional neural network⁴⁶ (CNN).

Second, we have comprehensively compared the performance of these methods with and without target labels. The results show that by having target labels for learning the target domain, the existing methods show high accuracy. However, the target labels in practice come at the cost of numerous workforce and energy. In the absence of target labels, the existing methods fail to provide a reliable estimation. By contrast, the proposed framework achieves accurate SOH estimation without target labels, reducing the MAE and maximum absolute error by more than half.

Third, three benchmarks are designed for ablation experiments to better verify the proposed framework. The excellent performance of our method can be attributed to the swarm-driven and domain adaptation strategies. To demonstrate this, ablation experiments are performed to verify the role of these strategies. Benchmark 1 and Benchmark 2 are created by disabling the swarm-driven and domain adaptation strategies of the proposed framework, respectively. Benchmark 3 is designed by disabling both strategies. Correspondingly, we added a table at the bottom of Fig. 4 to clearly show the focus of these benchmarks. The results show that, without the help of either of the two strategies, the estimation performance degrades significantly.

The reference to this response is as follows:

35. Richardson, R. R., Birkel, C. R., Osborne, M. A. & Howey, D. A. Gaussian Process Regression for in Situ Capacity Estimation of Lithium-Ion Batteries. *IEEE Trans Industr Inform* 15, 127–138 (2019).
44. Li, Y. et al. Random forest regression for online capacity estimation of lithium-ion batteries. *Appl Energy* 232, 197–210 (2018).
45. Guo, Y., Huang, K., Yu, X. & Wang, Y. State-of-health estimation for lithium-ion batteries based on historical dependency of charging data and ensemble SVR. *Electrochim Acta* 428, 140940 (2022).
46. Tian, J., Xiong, R., Shen, W., Lu, J. & Sun, F. Flexible battery state of health and state of charge estimation using partial charging data and deep learning. *Energy Storage Mater* 51, 372–381 (2022).

Comment 8: Results – Comparison with existing methods – In Figure 7, the authors show a very limited number of capacity predictions. From the figure, it is hard to get a sense of what “3%” error looks like on a real cell. The authors are advised to revise Figure 7 to include more prediction points or are suggested to create a new figure showing predicted values overlaid on a capacity-fade curve.

Thank you so much for your valuable comment.

We strongly agree with your statement. This figure in the revised manuscript is renamed to Fig. 6 (Page 20). It visualizes the feature explanation process of a DNN with and without domain adaptation and is used to verify the effectiveness of domain adaptation. As you suggested, all estimates of the DNN are included (indicated by blue dots in the sample-SOH curve) in the revised version. It is easy to get a sense of overestimation from the dataset #5 estimation results in Fig. 6a. This is because, without domain adaptation, the DNN shows abnormally high feature values for dataset #5. In contrast to Fig. 6a, the gaps in explanation maps between the two datasets are significantly mitigated by domain adaptation. Thanks to this, the DNN shows accurate estimations even in the absence of the dataset #5 labels (see Fig. 6b).

The comparison between the real and estimated values of SOH is shown in Fig. S4. (Supplementary Note 6). The abscissa in this figure is the real value of SOH, and the ordinate is the estimated value of SOH. The black line denotes the zero-error baseline. It is easy to get a sense of our accuracy by comparing the distance between the estimates and the black line. The result shows that the estimated value of the proposed framework is very close to the black line. It indicates that the proposed framework can provide accurate SOH estimation in the absence of target labels.

Comment 9: Results – Cross-manufacturer battery SOH estimation in the absence of target labels – The authors highlighted the importance of domain balance in estimating without target labels but didn’t clarify the effects of domain imbalance and trimming process in estimation performance. The reviewers suggest that the authors clarify the potential effects of skewed SOH distribution on the estimation performance and incorporate the techniques

for handling that in the framework.

We greatly appreciate your professional and insightful comment.

Following the suggestions, we have incorporated the trim into the proposed framework in the revised manuscript.

First, we have provided a detailed description of the trim in the “*Data processing*” section (“Methods” section, Pages 25-26). The trim strategy is mainly achieved in two steps. In the first step, the source domain samples are grouped into several bins with a uniform width (set to 2% in this work) according to their labels. The number of samples in each bin that need a trim is determined by minimizing the skewness of the new source domain and the number of discarded samples. In the second step, according to these calculation results, the samples in each bin are randomly discarded to generate a new source domain. No matter which dataset is used as the source domain, the goal of trimming is to generate a new source domain with a symmetrical distribution of samples by randomly discarding samples.

Second, we have added a description to clarify the effect of the trim (“Framework overview” section, Page 6, Line 15): “... We also design a trimming round to form a new source domain with a balanced SOH distribution by discarding some samples...”. We have also added a graphics module of the trim in Fig. 1a (“Framework overview” section, Page 8).

Third, we have clarified the effect of the skewed sample distribution in the source domain on SOH estimation (“*Cross-dataset battery SOH estimation in the absence of target labels*” section, Pages 11-13). In this section, we compare the performance of the proposed framework before and after the trim. The results show that the proposed framework is effective in estimation before the trim, and the overall accuracy is higher after the trim. In conclusion, the proposed framework can achieve accurate cross-dataset SOH estimation in the absence of target labels and can be improved after trimming the sample distribution in the source domain.

Comment 10: Conclusion – The authors conclude that “proposed framework is friendly for further applications.” This sentence is not clear and could be confusing for the readers. Please provide more details to clarify how multiple runs with various hyperparameters show the

friendliness of the framework for further applications.

Thank you so much for your professional comment.

To clarify this point, we have revised the section “*Performance over multiple independent runs with various hyper-parameters*” to “*Estimation performance with various hyper-parameters*” (Pages 21-22). In the revised section, we focus on investigating the impact of some crucial hyper-parameters on the estimation performance of the proposed framework, including the size of the DNN swarm, activation functions, number of channels and number of layers of the CNN. The results validate the hyper-parameter tuning of the framework and also provide some reference for the selection of hyperparameters:

(a) A size of 50 is sufficient for accurate estimation by suppressing the MAE below 2%.

(b) The ReLU activation function shows the highest accuracy and is therefore preferred when applying the framework.

(c) The number of channels less than 128 is sufficient to provide an accurate estimation.

(d) Multiple CNN layers are conducive to high accuracy.

Correspondingly, the description “the proposed framework in the “Conclusion” section is friendly for further applications” is changed to: “Finally, we investigate the impact of the crucial hyper-parameters on the estimation performance, and provide references for hyper-parameter selection to apply the proposed framework better...” (Page 23, Line 21).

Comment 11: Editorial Comments – The authors repeatedly misuse the word “herein” and other common conjoining words (nevertheless, however, etc.) in the paper. The authors are advised to avoid using these words unless necessary and to proofread their paper using advanced grammar-checking software (such as Grammarly) to further improve their paper’s sentence structure and overall clarity.

Thank you for your professional comments.

The revised manuscript has been revised to avoid using the mentioned words unless necessary. The full text has been further improved by using Grammarly.

Comment 12: Editorial Comments – The authors are misusing the word “rational” and its other forms (e.g., “rationalize”). Please revise sentences using this word. Consider replacing it and only using when absolutely necessary.

Thank you for your professional comments.

After the modification, the word “rational” and its other forms have almost been replaced. In the revised manuscript, the word “Rationalization” is only used in the title of the “*Rationalization of predictive performance*” section (Page 16).

Comment 13: Editorial Comments – Please check the grammar structure of the sentences for better readability in:

1) The Conclusion section, line 6.

Thank you for your professional comments.

We have changed the relevant sentence to: “This framework **selectively integrates** the estimations of a swarm of DNNs **into** a reliable SOH **estimate rather than relying on a single DNN**. **Each** DNN is trained for source labels and domain invariance of degradation features simultaneously...” (Page 23, Line 4)

2) The Conclusion section, line 22.

Thank you for your professional comments.

We have changed the relevant sentence to: “In summary, our work highlights the **potential** of deep learning in supplanting time- and resource-consuming **battery degradation tests...**” (Page 23, Line 24)

3) The Framework Overview section, line 17.

Thank you for your professional comments.

We have changed the relevant sentence to: “Each DNN in the proposed framework **has identical hyper-parameters** (Fig. 1b) but **is initialized with different random seeds based on the He initializer**³⁸. As the input of DNN, partial charging curves from both source and target domains are **first gridded with a voltage interval** of 10 mV to reduce the data burden...” (Page 6, Line 7)

Reviewer #2:

General comment: The manuscript tackles the lack of SOH labels for SOH estimation by a framework aggregating several deep learning networks.

We greatly appreciate your recognition of our contributions. Estimating battery SOH in the absence of target labels is challenging but of great significance for battery further development. In this manuscript, we focus on developing a target label-agnostic solution for SOH estimation. It is promising to help the development of battery management systems for new-generation batteries using only existing experimental data, saving considerable time and resources. It is also expected to motivate the utilization of large-scale field data without labels. We also appreciate your comments pointing out the shortcomings of our original manuscript. We have carefully considered all the comments and suggestions. The following are detailed responses to all the comments.

- 1) The approach seems convincing from a theoretical point of view. My major concern is that the hyperparameter tuning of the DNNs is insufficient (not well documented & only few trials).

Thank you for pointing out the shortcomings of our study. Hyper-parameters are difficult to optimize, especially without target labels. We also greatly appreciate your suggestions for improvement.

Following the suggestions, we have revised the section “*Performance over multiple independent runs with various hyper-parameters*” to “*Estimation performance with various hyper-parameters*” (Pages 21-22). In the revised section, we have made more trials to investigate the impact of some crucial hyper-parameters on the estimation performance of our framework, including the size of the DNN swarm, activation functions, number of channels and number of layers of the CNN. The results validate the hyper-parameter tuning of the framework and also provide some reference for the selection of hyperparameters:

- (a) A size of 50 is sufficient for accurate estimation by suppressing the MAE below 2%.
- (b) The ReLU activation function shows the highest accuracy and is therefore preferred when applying the framework.
- (c) The number of channels less than 128 is sufficient to provide an accurate estimation.

(d) Multiple CNN layers are conducive to high accuracy.

Thus, the analysis of hyper-parameters is strengthened in the revised manuscript.

- 2) Also, the swarm approach is interesting, but a benchmark to a single DNN with a similar number of parameters like the whole DNN swarm is missing.

We greatly appreciate your professional and insightful comment.

Following the suggestions, we have designed a benchmark using a single DNN with a comparable number of parameters to the DNN swarm in the “*Comparison with existing methods*” section (Pages 15-16). This benchmark, named Benchmark 1 in the revised manuscript, is equivalent to disabling the swarm-driven strategy of the proposed framework. Therefore, it can be used to verify the effectiveness of the swarm-driven strategy. The results show that without the help of the swarm-driven strategy, the estimation accuracy deteriorates significantly.

Inspired by the suggestions, we have also designed ablation experiments in the revised manuscript. This benchmark is treated as one of the benchmarks in the ablation experiments. In addition to this, there are two benchmarks. One is created by disabling the swarm-driven and domain adaptation strategies, and the other is created by disabling the domain adaptation strategy. To clearly show the focus of these benchmarks, we added a table at the bottom of Fig. 4. The settings for these benchmarks can be found in Supplementary Note 7. The comparison results show that, without the help of either of the two strategies, the estimation performance degrades significantly as expected. The detailed estimation results of these benchmarks can be found in Supplementary Note 8.

In general, the performance comparison in the revised manuscript is strengthened.

- 3) In the section “Comparison with existing methods” no existing methods are used for comparison, only benchmarks defined by the authors are defined. This hinders comparability to state of the art work. It would be interesting to compare models with and without the absence of the target labels (i.e. how much worse does the SOH estimation get with absence of the target labels compared to available target labels? How much better is the SOH estimation with absence of the target labels and your approach?).

We greatly appreciate your professional and valuable comments.

To do this, we have revised the “*Comparison with existing methods*” section (Page 15). Compared with the previous version, the revised version mainly focuses on the following changes.

First, to enhance comparability, four popular methods from the literature are employed as representatives of existing methods, including Gaussian process regression³⁵ (GPR), random forest⁴⁴ (RF), support vector regression⁴⁵ (SVR), and convolutional neural network⁴⁶ (CNN).

Second, following the suggestions, we have comprehensively compared the performance of these methods with and without target labels. The results show that by having target labels for learning the target domain, the existing methods show high accuracy. However, the target labels in practice come at the cost of numerous workforce and energy. In the absence of target labels, the existing methods fail to provide a reliable estimation. By contrast, the proposed framework achieves accurate SOH estimation without target labels, reducing the MAE and maximum absolute error by more than half.

The reference to this response is as follows:

33. Richardson, R. R., Birkl, C. R., Osborne, M. A. & Howey, D. A. Gaussian Process Regression for in Situ Capacity Estimation of Lithium-Ion Batteries. *IEEE Trans Industr Inform* 15, 127–138 (2019).
 44. Li, Y. et al. Random forest regression for online capacity estimation of lithium-ion batteries. *Appl Energy* 232, 197–210 (2018).
 45. Guo, Y., Huang, K., Yu, X. & Wang, Y. State-of-health estimation for lithium-ion batteries based on historical dependency of charging data and ensemble SVR. *Electrochim Acta* 428, 140940 (2022).
 46. Tian, J., Xiong, R., Shen, W., Lu, J. & Sun, F. Flexible battery state of health and state of charge estimation using partial charging data and deep learning. *Energy Storage Mater* 51, 372–381 (2022).
- 4) It would be handy to have linked cross-reference in the document to Figures, Sections, etc. Some sentences are quite long & nested and, thus, hard to understand. Figures should be vectorized (e.g. Fig 3 is not).

Thank you so much for your professional comments.

To solve the mentioned problems, we have made the following modifications:

First, we have linked cross-references in the revised manuscript to the Figures, Tables, and Sections.

Second, most long sentences have been rephrased for better understanding.

Third, all the figures have been vectorized in the revised manuscript.

Comment 1: Abstract

- 1) “the time-consuming experiments” are mentioned twice but have never been mentioned before. In what respect and why are they time-consuming? Are they maybe also expensive?

We greatly appreciate the constructive comments and suggestions.

First, to avoid duplicate mentions, we have removed the second mention of “the time-consuming experiments”: “...This work emphasizes the power of deep learning in precluding **degradation** experiments and highlights the promise of rapid development of battery management algorithms for new-generation batteries using only previous experimental data.” (“Abstract” section, Page 2, Line 11)

Second, following the suggestions and taking the employed datasets as examples, we have employed the current profile and capacity trajectories to estimate the minimum additional experimental time (neglecting the rest time during the battery degradation tests) in Supplementary Note 2. The estimated test duration of the datasets is listed in Table 1 (“*Data generation*” section, Page). The results show that a degradation experiment requires 644-8473 hours to reduce the battery SOH to 75%, so it is time-consuming. It is also found that a degradation experiment covers at least hundreds of thousands of charge and discharge cycles, consuming a lot of energy. So, degradation experiments are not only time-consuming but also expensive or resource-consuming. Thus, we have revised the first mention of “the time-consuming degradation experiments” to “the time- and resource-consuming degradation experiments”: “...While accurate SOH estimation has progressed markedly, the time- **and resource-consuming degradation** experiments hinder **the** development of SOH estimation methods...” (“Abstract” section, Page 2, Line 2)

- 2) “deep learning networks” is not a common term. Do you mean deep neural networks or other deep learning models? Please be more specific.

Thank you so much for kindly pointing out this error.

We have changed the “deep learning networks” to “deep neural networks” in the “Abstract” section: “...This framework **integrates** a swarm of deep **neural** networks equipped with **domain adaptation** to produce accurate estimation...” (“Abstract” section, Page 2, Line 6)

3) Does “absolute error” refer to MAE?

Thank you so much for your professional comments.

Here, the “absolute error” does not refer to mean absolute error (MAE). It indicates the absolute value of the difference between the measured SOH and true SOH. Given a SOH y and its estimate \hat{y} , the “absolute error” is defined as $AE = |y - \hat{y}|$.

Both “absolute error” and MAE are used in this study to evaluate the estimation performance comprehensively. To clarify this, they are formulated in the “*SOH estimation with trained DNNs*” section (Page 30, Equation 11). The validation results show that the proposed framework can ensure absolute errors of less than 3% for 89.4% of samples (less than 5% for 98.9% of samples), with a maximum absolute error of less than 8.87% in the absence of target labels (“Abstract” section, Page 2, Line 9). And its MAE is within 1.43% (“*Comparison with existing methods*” section, Page 15, Line 17).

4) “within 3% for up to 88.6% of the cycles”: in the Intro this is formulate more precisely and understandably (“framework can achieve absolute errors of 3% for 88.6% of samples”). It seems that the results are little cherry-picked. At least you should give information about the remaining 11.4% of the samples. How bad is the prediction for those? Why is the prediction performance of those samples not relevant?

Thank you so much for pointing out the shortcomings. We also greatly appreciate your suggestions for improvement.

First, we have changed the word “cycles” to “samples” (“Abstract” section, Page 2, Line 9) for a more precise expression.

Second, following the suggestions, we describe the error for 100% of the samples by modifying: “...The validation results indicate that the proposed framework can ensure absolute errors of less than 3% for 89.4% of samples (less than 5% for 98.9% of samples), with a maximum absolute error of less than 8.87% in the absence of target labels...”. (“Abstract” section, Page 2, Line 9)

Third, we visualize the error distribution as a function of true SOH in Fig. 3b (“*Cross-dataset battery SOH estimation in the absence of target labels*” section, Page 2, Line 9). The result shows that the high-error samples mainly belong to the high-SOH samples (SOH > 95%) and are due to domain imbalance. Nevertheless, their proportion is very low. From the

overall sample point of view, the proposed framework can make accurate estimations in the absence of target labels.

Comment 2: Intro

- 1) “In general, SOH is defined as the ratio between the present capacity to the initial maximum capacity”: In general the SOH is not defined capacity-based, but in most cases! There are other ways to define the SOH, e.g., referring to internal resistance, number of charge/discharge cycles that you only mention later in “Limitations and outlook”. The intro is a good spot to already do that and argue, why you opt for the capacity-based SOH.

We greatly appreciate your professional and valuable comments.

As you mentioned, battery SOH has been defined in various forms. It can be defined by the service time (i.e., the number of charge/discharge cycles) or by the increase in the internal resistance. Capacity-based SOH is used in most cases but is not the only definition. To clarify this, we have revised the relevant description as: “Battery SOH has been defined in various forms. It can be defined by the service time⁹ or by the increase in the internal resistance¹⁰. Although these variables are easily measurable, battery degradation is also accompanied by capacity loss, whose accurate determination impacts other battery management tasks such as driving range estimation and life prediction. Thus, SOH defined as the ratio between the present capacity and the initial capacity is drawing broad attention^{11,12}. However, the capacity measurement requires...” (“Introduction” section, Page 3, Line 11)

- 2) “Standard SOH calibration requires fully charging or discharging a battery between the voltage limits”: this is not sufficient. Also, temperature and current profile have an influence. Simply charging or discharging until the voltage limits is not sufficient.

Thank you so much for your professional comments and suggestions.

Following the suggestions and to express more precisely, we have changed the imprecise expression “between the voltage limits” to “with specific protocols¹³”. Literature [13] is cited to explain the “specific protocols”. The relevant description (i.e., the second paragraph in the

“Introduction”) is as follows: “...However, the capacity measurement requires completely charging or discharging the batteries with specific protocols¹³, which is not practical for batteries in use...” (“Introduction” section, Page 3, Line 16)

The reference to this response is as follows:

13. Xiong, R., Li, L. & Tian, J. Towards a smarter battery management system: A critical review on battery state of health monitoring methods. *J Power Sources* 405, 18–29 (2018).

3) “A growing number of literatures”: “literature” is a singular-only noun

Thank you so much for kindly pointing out this error.

We have corrected this expression to “A growing body of literature” in the revised manuscript: “A growing **body of literature**^{26,27} **applies** retraining or fine-tuning techniques for **SOH estimation of** various types of batteries....” (“Introduction” section, Page 4, Line 10)

Comment 3: Framework overview:

1) “driven by a dropout rule”: without a look at the figure it is unclear that the dropout rule does not refer to single neurons, but to complete DNNs. Why did you opt against a single DNN with neuron-wise dropout during training?

Thank you so much for kindly pointing out this error.

First, to avoid confusion, we have changed the description “dropout rule” to “selection”. The neuron-wise dropout is used to avoid overfitting, which means that some nodes are randomly “dropped out” of the network at the training stage, but all nodes participate in the application. Unlike this dropout, all DNNs in the proposed framework are trained but only a swarm of them are selected to participate in the estimation. Thus, the framework selectively integrates the estimates of a swarm of DNNs into a reliable SOH estimate rather than relying on a single DNN. The relevant description is revised as: “... This framework is designed for reliable estimation by **selectively integrating** the estimates from multiple DNNs...” (“Framework overview” section, Page 6, Line 2)

Second, we have demonstrated that relying only on a single DNN may yield unreliable SOH estimates in the absence of target labels in the “*Rationalization of predictive performance*” section (Pages 16-18). Beyond that, benchmark 1 (i.e., a single DNN with a

comparable number of parameters to the DNN swarm) can also demonstrate the unreliable performance of a single DNN (“*Comparison with existing methods*” section, Pages 15-16). Thus, our framework selectively integrates the estimates of a swarm of DNNs into a reliable SOH estimate rather than relying on a single DNN.

- 2) “The current is scaled to C-rate to cope with inconsistent operating current range among various types of LIBs.”: Using the C-rate makes sense. But what do you exactly mean by “inconsistent operating current range”? Please provide an example.

Thank you so much for pointing out this issue.

We agree that the expression “inconsistent operating current range” is confusing. In fact, what we are expressing is that the source domain battery and the target domain battery are generally inconsistent in nominal capacity. Thus, for a clearer description, we have changed the relevant sentences to: “...Before sub-trainings, partial charging curves from both source and target domains are normalized by their nominal capacity...” (“Framework overview” section, Page 6, Line 11)

- 3) “Each DNN in the proposed framework is structurally identical (Fig. 1b) but independently trained with different initialized parameters”: Do you mean the hyperparameters are identical (no. layers, no. neurons), but the weights and biases are initialized differently? How are they initialized: Same method (e.g. He, Glorot), but different random seed?

We greatly appreciate your professional comments.

Specifically, each DNN in the proposed framework has identical hyper-parameters but is initialized with different random seeds based on the He initializer. To clarify this, we have revised the relevant sentence as: “...Each DNN in the proposed framework has identical hyper-parameters (Fig. 1b) but is initialized with different random seeds based on the He initializer³⁸...” (“Framework overview” section, Page 6, Line 18)

The work by He et al. [38] is cited to further illustrate the He initializer:

38. He, K., Zhang, X., Ren, S. & Sun, J. Delving Deep into Rectifiers: Surpassing Human-Level Performance on ImageNet Classification. in 2015 IEEE International Conference on Computer Vision (ICCV) 1026–1034 (IEEE, 2015). doi:10.1109/ICCV.2015.123.

- 4) “respectively” mean “in the order mentioned” – but the order of the domains/branches does not in this case.

Thank you so much for kindly pointing out this error.

To avoid using the words “respectively” and “branches”, we have revised the description as: “...After that, **feature vectors** of the source domain are flattened and fed into a terminal **fully connected (TFC) layer** to generate their SOH estimates. **These estimates are used together with the source domain labels to calculate the source domain loss. On the other hand, feature vectors** of **target-domain samples** are flattened to a middle **fully connected (MFC) layer** for reconstructing **their feature vectors**. **These reconstructed feature vectors play two roles. The first is to quantify the domain gap together with the source domain feature vectors. The second is to provide estimates of target domain samples (treated as the pre-estimates of each trained DNN) in the estimation procedure, where the reconstructed feature vectors are further fed into the same TFC as the source domain for regression...**” (“Framework overview” section, Page 6, Line 22)

Furthermore, we have modified Fig. 1b by using arrows of two different colors to distinguish the sample paths of the two domains. (Page 8)

- 5) “terminal fully connection layer (TFC)” should be “terminal fully connection (TFC) layer” because “layer” is not part of the acronym. Same for MFC

Thank you so much for kindly pointing out this error.

The descriptions of the “terminal fully connection layer (TFC)” are all changed to “terminal fully connected (TFC) layer”. Also, the descriptions of the “middle fully connected layer (MFC)” are all changed to “middle fully connected (MFC) layer” in the revised manuscript.

- 6) “firstly” -> “first”

Thank you so much for your kind reminder.

The word “firstly” is all changed to “first” in the revised manuscript.

Comment 4: Data Generation

- 1) In Table 1: “2.7~4.2”: Tilde should be “-”

Thank you so much for your kind reminder.

We have revised Table 1 according to your suggestions. (Page 9)

- 2) Why are you only considered each one cell from the five data sets? I assume even within each of the data sets there is intrinsic cell-to-cell variability from production, but also due to different operational protocols. How could this be considered?

We greatly appreciate your rigorous and professional comment.

First, to avoid misleading, the word “cell” is changed to “dataset” in the revised manuscript. This is because each dataset in this work contains several cells. The cross-dataset SOH estimation focuses on “cells-to-cells” estimation but not “cell-to-cell” estimation.

Second, as you mentioned, we do not consider cell-to-cell variability but focus on dataset-to-dataset variability. To our knowledge, SOH estimation in target label-agnostic cases generally spans different factors such as applications, manufacturers and chemistries. Five datasets are employed in this manuscript to reflect such situations, as they differ in these factors. We think that “cross-dataset” is a better match for these situations, so the previous words “cross-manufacturer” are changed to “cross-dataset” in the revised manuscript. Furthermore, our framework is shown to achieve accurate cross-dataset SOH estimation in the absence of target labels. Thus, by changing the cross-dataset domain to the cross-cell domain, we believe that the proposed framework can also achieve accurate estimation.

- 3) Fig 2. “SOH distributions”: I have never seen this type of SOH-residence-based visualization for SOH trajectories. I see why you opt for this type of visualization in contrast to the common “cycle vs. SOH” plots given your further plots, e.g. in Fig. 3 c) & d). I would appreciate a little explanation of how to interpret this plot (e.g. sth. like binning/ buckets of SOH values)

Thank you so much for your kind reminder.

As you mentioned, this type of visualization is more suitable for our focus. To avoid misleading, we have changed the word “cycles” to “samples” in Fig. 2 and rewritten the caption to better clarify this: “Fig. 2. Lifelong charging curve family and histogram of the

SOH of the selected five types of LIBs produced by five manufacturers. (a) #1, (b) #2, (c) #3, (d) #4, (e) #5. **The histograms in subplots show the number of samples with different SOH.**”
(Page 10)

Comment 5: Cross manufacturer battery SOH estimation ...

1) Fig. 3 a): y-axis label “Error” is not precise – which error do you use exactly? RMSE, MSE, MAE, MAPE,...? This does also apply to the other figures.

Thank you so much for your rigorous comment.

We use “absolute error” to describe the estimation performance in Fig. 3. Following the suggestions, we have changed the y-axis label “error” to “absolute error” in Fig. 3 (Page 14). In addition, terms related to “error” in all figures have been updated in the revised manuscript according to the type of error.

2) “within 4.90% for the lower SOH bounds” & “within 5.01% even when the domains are extremely imbalanced (i.e. 95% lower SOH bound)”: Where exactly is that visible in Fig. 3? I see errors up to 10 %, e.g. source domain #4 & target domain # 1.

Thank you so much for your rigorous comment.

This confusion is mainly due to the confusing terms we use for evaluating errors in the original manuscript. To clarify this, we have modified Fig. 3 to better distinguish between “absolute error” and MAE (Page 14). The y-axis label “error” is changed to “absolute error”. The black horizontal lines are employed to represent MAEs. From the revised figure, it is easier to see that the MAE (see the black horizontal line) in all cases is within 5.01% before the trim.

3) “In other words, cases with a lower SOH bound of 95%”: Isn’t 95% a higher bound than e.g. 80%? (95 > 80%)? This is already confusing in previous sentences.

Thank you so much for kindly pointing out this error.

It is the truth that 95% is greater than 80%, and what we want to express is that a 100-95% distribution has a narrower range than a 100-80% distribution. To avoid confusion, the SOH of batteries in the target domain distributed from 100% to 95%, 90%, 85%, 80% are denoted

as the cases of “~95%”, “~90%”, “~85%”, “~80%”, respectively. (“*Cross-dataset battery SOH estimation in the absence of target labels*” section, Page 11, Line 9). The adjectives “high” and “low” are no longer used to describe these situations in the revised manuscript.

- 4) How do you trim the dataset of cell #2? This is a very interesting point because battery datasets will mostly be imbalanced so dealing with that is also relevant for other models & approaches. How do you make sure you don’t cherry-pick?

We greatly appreciate your insightful comments and suggestions.

Following the suggestions, we have incorporated the trim into the proposed framework in the revised manuscript.

First, we have provided a detailed description of the trim in the “*Data processing*” section (“*Methods*” section, Pages 25-26). The trim is mainly achieved in two steps. In the first step, the source domain samples are grouped into several bins with a uniform width (set to 2% in this work) according to their labels. The number of samples in each bin that need a trim is determined by minimizing the skewness of the new source domain and the number of discarded samples. In the second step, according to these calculation results, the samples in each bin are randomly discarded to generate a new source domain. No matter which dataset is used as the source domain, the goal of trimming is to generate a new source domain with a symmetrical distribution of samples by randomly discarding samples. This guarantees no cherry-picking.

Second, we have added a description to clarify the effect of the trim (“*Framework overview*” section, Page 6, Line 15): “... We also design a trimming round to form a new source domain with a balanced SOH distribution by discarding some samples...”. Correspondingly, we have also added a graphics module of the trim in Fig. 1a (“*Framework overview*” section, Page 8).

Third, we have clarified the effect of the skewed sample distribution in the source domain on SOH estimation (“*Cross-dataset battery SOH estimation in the absence of target labels*” section, Pages 11-13). In this section, we compare the performance of the proposed framework before and after the trim. The results show that the proposed framework is effective in estimation before the trim, and the overall accuracy is higher after the trim. Thus, the proposed framework can achieve accurate cross-dataset SOH estimation in the absence

of target labels and can be improved after trimming the sample distribution in the source domain.

- 5) “up to 88.6% cycles with absolute error less than 3%”: This is very confusing wording to me because it may suggest you are having “cycles” as an output. What does “88.6% cycles” mean? I interpret it refers to 88.6% of the cycles in the dataset, having one sample per cycle.

Thank you so much for your kind reminder.

We agree that the term “cycles” is confusing in this manuscript. We focus on SOH estimation which aims to estimate SOH from battery operating signals. In this regard, each cycle in the battery degradation test can provide a sample for SOH estimation. To avoid confusion, we have replaced the term “cycles” in the text with “samples”.

Comment 6: Comparison with existing methods

- 1) The title of this section is misleading because no “existing methods” are cited or mentioned in this section. Only benchmarks proposed by the authors of this work are evaluated. This limits comparability to other work.

Thank you so much for your professional comments and suggestions.

Following the suggestions, we have revised the “*Comparison with existing methods*” section (Page 15). Compared with the previous version, the revised version mainly focuses on the following changes.

First, we have clarified the background of the methods involved in the comparison. Four popular methods from the literature are employed as representatives of existing methods, including Gaussian process regression³⁵ (GPR), random forest⁴⁴ (RF), support vector regression⁴⁵ (SVR), and convolutional neural network⁴⁶ (CNN).

Second, we have comprehensively compared the performance of these methods with and without target labels. The results show that by having target labels for learning the target domain, the existing methods show high accuracy. However, the target labels in practice come at the cost of numerous workforce and energy. In the absence of target labels, the

existing methods fail to provide a reliable estimation. By contrast, the proposed framework achieves accurate SOH estimation without target labels, reducing the MAE and maximum absolute error by more than half.

Third, three benchmarks are designed for ablation experiments to better verify the proposed framework. The excellent performance of our method can be attributed to the swarm-driven and domain adaptation strategies. To demonstrate this, ablation experiments are performed to verify the role of these strategies. Benchmark 1 and Benchmark 2 are created by disabling the swarm-driven and domain adaptation strategies of the proposed framework, respectively. Benchmark 3 is designed by disabling both strategies. Correspondingly, we added a table at the bottom of Fig. 4 to clearly show the focus of these benchmarks. The results show that, without the help of either of the two strategies, the estimation performance degrades significantly.

The reference to this response is as follows:

35. Richardson, R. R., Birkl, C. R., Osborne, M. A. & Howey, D. A. Gaussian Process Regression for in Situ Capacity Estimation of Lithium-Ion Batteries. *IEEE Trans Industr Inform* 15, 127–138 (2019).
 44. Li, Y. et al. Random forest regression for online capacity estimation of lithium-ion batteries. *Appl Energy* 232, 197–210 (2018).
 45. Guo, Y., Huang, K., Yu, X. & Wang, Y. State-of-health estimation for lithium-ion batteries based on historical dependency of charging data and ensemble SVR. *Electrochim Acta* 428, 140940 (2022).
 46. Tian, J., Xiong, R., Shen, W., Lu, J. & Sun, F. Flexible battery state of health and state of charge estimation using partial charging data and deep learning. *Energy Storage Mater* 51, 372–381 (2022).
- 2) “additionally treats the initial cycle labels of the target domain as 100%”: As 100% of what? Are you assuming 100% SOH for all initial cycles in the target domain?

Thank you so much for kindly pointing out this issue.

In this work, SOH is defined as the ratio between the present capacity and the initial capacity. On this basis, the label of the initial cycle for any battery is 100%. To avoid confusion, we have removed this statement from the manuscript. The benchmarks are distinguished by ablation experiments in the revised version. To clearly show the focus of these benchmarks, we added a table at the bottom of Fig. 4 (Page 16). It is easy to see that Benchmark 1 and Benchmark 2 are created by disabling the swarm-driven and domain adaptation strategies of the proposed framework, respectively. Benchmark 3 is designed by

disabling both strategies. With these benchmarks for comparison, we can better validate the proposed framework.

3) “Thus in the label-agnostic”: Comma missing after “thus”

Thank you so much for kindly pointing out this error. We have corrected all such errors in the revised manuscript.

Comment 7: Performance over multiple independent runs with various hyper-parameters

1) “no feedback from the target domain is available for refining these hyper-parameters.”:

That is indeed true. Potentially, if you had feedback, e.g. by assuming SOH = 100 & at the first cycle as you do in your benchmarks: How would you change the number of layers, number of neurons, kernel size, etc. while doing the transfer learning? I am not aware of any approaches accomplishing that. Thus, I am curious how you would accomplish that. If you are not considering that I suggest a rephrasing.

We greatly appreciate your rigorous and professional comment. It is the truth that hyper-parameters are difficult to optimize, particularly in the absence of target labels.

Following the suggestions, we have rephrased the section “*Performance over multiple independent runs with various hyper-parameters*” as “*Estimation performance with various hyper-parameters*” (Page 21). In the revised version, we have made more trials to investigate the impact of some crucial hyper-parameters on the estimation performance of the proposed framework, including the size of the DNN swarm, activation functions, number of channels and number of layers of the CNN. The results validate the hyper-parameter tuning of the framework and also provide some reference for the selection of hyperparameters:

(a) A size of 50 is sufficient for accurate estimation by suppressing the MAE below 2%.

(b) The ReLU activation function shows the highest accuracy and is therefore preferred when applying the framework.

(c) The number of channels less than 128 is sufficient to provide an accurate estimation.

(d) Multiple CNN layers are conducive to high accuracy.

In general, we have strengthened the discussion of hyper-parameter settings in the revised

manuscript.

- 2) “Herein we do not discuss”: Comma missing after “herein”

Thank you so much for kindly pointing out this error. We have corrected all such errors in the revised manuscript.

- 3) “Instead, we conduct multiple independent runs with various hyperparameters to verify the sensitivity of the proposed framework to the hyper-parameter settings”: Your hyperparameter tuning with “additional six groups of randomly formulated hyperparameters” does not convince me regarding optimality of the hyperparameter choice. What is the range of number of layer, number of neurons, activation function, regularization/dropout/early stopping you have considered? Furthermore, six random trials seems very little to me. I would prefer using some established Bayesian optimization approach for hyperparameter tuning. Or also more random trials.

We greatly appreciate your professional comment.

Since hyper-parameters are hard to optimize without target labels in practice, we have conducted more trials to analyze the impact of some crucial hyper-parameters on estimation performance in the revised manuscript. The section “*Performance over multiple independent runs with various hyper-parameters*” is rephrased as “*Estimation performance with various hyper-parameters*”. In this section, we further investigate the impact of some crucial hyper-parameters on the estimation performance of the proposed framework, including the size of the DNN swarm, activation functions, number of channels and number of layers of the CNN.

First, the size of the DNN swarm is set to 1, 50, 100, ..., and 300, respectively, to study its impact. The results show that a size of 50 is sufficient for accurate estimation. One can balance the accuracy and computational cost by tuning the swarm size in practice.

Second, we examine the influence of activation functions by comparing the estimation performance using ReLU, Tanh, Sigmoid, and LogSigmoid, respectively. The results show that the ReLU activation function shows the highest accuracy and is preferred when applying the framework.

Third, we span the number of CNN layers from 1 to 4, and the number of channels for all layers is assumed to be identical and belongs to [32, 64, 128, 256]. The results show that

increasing the number of channels does not always reduce the MAE except for the 1-layer CNN. On the other hand, multiple CNN layers are conducive to high accuracy. One might need to find a suitable number of CNN layers to balance the estimation accuracy and the computational cost.

Comment 8: Appendix/ Methods:

1) Putting the equations of Adam is not necessary or interesting for any reader in my opinion. Adam is a well-established, standard algorithm so referring to the initial, proposing paper is sufficient.

Thank you so much for your professional comments.

Following the suggestions, we have removed the equations of Adam in the revised manuscript and have indexed the literature that initially proposed the Adam algorithm: “... The widely-used Adam algorithm⁴⁷ is employed to optimize the parameters iteratively...”. (“Methods”, Page 28, Line 17)

The reference to this response is as follows:

47. Kingma, D. P. & Ba, J. Adam: A Method for Stochastic Optimization. 3rd International Conference on Learning Representations, ICLR 2015 - Conference Track Proceedings 1–15 (2014).

REVIEWERS' COMMENTS

Reviewer #1 (Remarks to the Author):

All my comments have been properly addressed. Thanks to the authors for their effort. I don't have any further concerns.

Reviewer #2 (Remarks to the Author):

Thanks for the extension description of the changes made to the manuscript. You have achieved a great improvement. The manuscript is now more clear.